

**Effect of tropical cyclones on the Stratosphere-Troposphere Exchange**

**observed using satellite observations over north Indian Ocean**

M. Venkat Ratnam[1*], S. Ravindra Babu[2], Siddarth Shankar Das[3], Ghouse Basha[1],

B.V. Krishnamurthy[4] and B.Venkateswararao[2]

[1]National Atmospheric Research Laboratory (NARL), Gadanki, India.

[2]Jawaharlal Nehru Technological University, Hyderabad, India.

[3]Space Physics Laboratory (SPL), VSSC, Trivandrum, India.

[4]CEBROSS, Chennai, India.

*vratnam@narl.gov.in , 08585-272123 (phone), 08585-272018 (Fax)

**Abstract**

Tropical cyclones play an important role in modifying the tropopause structure and dynamics as well as stratosphere-troposphere exchange (STE) processes in the Upper Troposphere and Lower Stratosphere (UTLS) region. In the present study, the impact of cyclones that occurred over the North Indian Ocean during 2007-2013 on the STE processes is quantified using satellite observations. Tropopause characteristics during cyclones are obtained from the Global Positioning System (GPS) Radio Occultation (RO) measurements and ozone and water vapor concentrations in the UTLS region are obtained from Aura-Microwave Limb Sounder (MLS) satellite observations. The effect of cyclones on the tropopause parameters is observed to be more prominent within 500 km from the centre of the tropical cyclone. In our earlier study, we have observed decrease (increase) in the tropopause altitude (temperature) up to 0.6 km (3K) and the convective outflow level increased up to 2 km. This change leads to a total increase in the tropical tropopause layer (TTL) thickness of 3 km within the 500 km from the centre of cyclone. Interestingly, an enhancement in the ozone mixing ratio in the upper troposphere is clearly noticed within 500 km from cyclone centre, whereas the enhancement in the water vapor in the lower stratosphere is more significant on south-east side extending from 500-1000 km away from the cyclone centre. The cross-tropopause mass flux for different intensities of cyclones are estimated and found that the mean flux from the stratosphere to the troposphere for cyclonic storms is $0.05 \pm 0.29 \times 10^{-3}$ kgm$^{-2}$ and for very severe cyclonic storms it is $0.5 \pm 1.07 \times 10^{-3}$ kgm$^{-2}$. More downward flux is noticed in the north-west and south-west side of the cyclone centre. These results indicate that the cyclones have significant impact in effecting the tropopause structure, ozone and water vapor budget and consequentially the STE in the UTLS region.

(**Keywords:** Tropical cyclone, tropopause, ozone, water vapor, STE processes.)

## 1. Introduction

The tropical cyclones with deep convective synoptic scale systems persisting for a few days to week, play an important role on the mass exchange between the troposphere and the stratosphere, and vice versa (Merril, 1998; Emmanuel, 2005). They transport large amount of water vapor, energy and momentum to the upper troposphere and lower stratosphere (UTLS) region (Ray and Rosenlof, 2007). Cyclones provide favorable conditions for entry of the water vapor-rich and ozone-poor air from surface to the lower stratosphere (LS) and dry and ozone-rich air from the LS to the upper troposphere (UT) leading to the stratosphere-troposphere exchange (STE) (Romps and Kuang 2009; Zhan and Wang, 2012; Vogel et al., 2014). These exchanges occur mainly around the tropopause and change the thermal and chemical structure of the UTLS region. The concentration of the water vapor transported from troposphere to stratosphere is controlled by the cold temperatures present at the tropopause and this is a major factor in the STE (Fueglistaler et al., 2009). As a consequence, the STE events play an important role in controlling the ozone in the UTLS region, which will affect the radiation budget of the Earth atmosphere (Intergovernmental Panel on Climate Change, 1996).

Water vapor has major consequences for the radiative balance and heat transport in the atmosphere. Enhanced ozone loss is a secondary effect of increasing water vapor.(Rind and Lonergan, 1995; Forster and Shine, 1999; Dvortsov and Solomon, 2001; Forster and Shine, 2002; Myhre et al., 2007; Intergovernmental Panel on Climate Change, 2007). Even very small changes in lower stratospheric water vapor could affect the surface climate (Riese et al., 2012). Soloman et al. (2010) reported the role of stratospheric water vapor in the global warming.LS water vapor plays an important role on the distribution of ozone in the lower stratosphere (Shindell, 2001). It is important contributor for long-term change in the LS temperatures (Maycock et al., 2014).

In general, most of the air enters into the stratosphere over the tropics (Brewer, 1949;
Dobson, 1956). As suggested by Newell and Gould-Stewart (1981), Bay-of-Bengal (BoB) is
one of the active regions where troposphere air enters into the stratosphere. It is also one of
the active regions for the formation of deep convection associated cyclones which contains
strong updrafts. Earlier studies have shown a close relationship between cyclones and
moistening of the upper troposphere (Wang et al., 1995; Su et al., 2006; Ray and Rosenlof,

78    2007).

Several studies have been carried out related to water vapor, ozone transport as well
as STE processes around the UTLS region during cyclones. Koteswaram (1967) described
the thermal and wind structure of cyclones in the UTLS region with the major findings of
cold core persisting just above the 15 km and the outflow jets very close to the tropopause.
Penn (1965) reported enchantment in ozone and warmer air situated above the tropopause
over the eye region during hurricane Ginny. Danielsen (1993) reported on troposphere-
stratosphere transport and dehydration in the lower tropical stratosphere during cyclone
period. Baray et al. (1999) studied the STE during cyclone Marlene and they observed
maximum of ozone change at 300 hPa level. Zou and Wu (2005) observed the variations of
columnar ozone in different stages of hurricane by using satellite measurements. Bellevue et
al. (2007) observed increase in ozone concentration in the upper troposphere during Tropical
Cyclone (TC) event. Significant contribution of cyclones on hydration of the UT is reported
by Ray and Rosenlof (2007) and injection of tropospheric air into the low stratosphere due to
overshooting convection by cyclones is reported by Romps and Kuang (2009). Das (2009)
and Das et al. (2016) have studied the stratospheric intrusion into troposphere during the
passage of cyclone by using Mesosphere-Stratosphere-Troposphere (MST) Radar
observations. Strong enhancement of ozone in the upper troposphere is observed during TCs
over BoB (Fadnavis et al., 2011). The increased ozone  levels in the boundary layer as well as

near surface by as much as 20 to 30 ppbv due to strong downward transport of ozone in the tropical convection is also observed (Betts et al., 2002; Sahu and Lal, 2006; Grant et al., 2008). Cairo et al. (2008) reported that the colder temperatures are observed in the Tropical Tropopause Layer (TTL) region during cyclone Davina and also reported on the impact of the TCs on the UTLS structure and dynamics at the regional scales. A detailed review on the effect of TCs on the UTLS can be found in same report. Recently, Ravindra Babu et al. (2015) reported the effect of cyclones on the tropical tropopause parameters using temperature profile obtained from Constellation Observing System for Meteorology, Ionosphere and Climate (COSMIC) Global Position System Radio Occultation (GPS-RO) measurements. Many studies have been carried out on the role of extra tropical cyclones on the STE (for example Reutter et al., 2015 and references therein) though the quantitative estimates of STE provided by these case studies varied considerably. However, the vertical and horizontal variation of ozone and water vapor in the UTLS region and cross-tropopause flux quantification during cyclones over north Indian Ocean is not well investigated.

In the present study, we investigate the spatial and vertical variations of ozone and water vapor in the UTLS region for all the cyclones occurred over north Indian Ocean during 2007 to 2013 by using Aura-Microwave Limb Sounder (MLS) satellite observations. The effect of cyclones on the tropopause characteristics is also presented using COSMIC GPS-RO measurements. We also present the cross-tropopause mass flux estimated for each of the cyclones.

**2. Data and Methodology**

In the present study, we used Aura-MLS water vapor and ozone measurements (version 3.3) provided by the Jet Propulsion Laboratory (JPL). The version 3.3 was released in January 2011 and this updated version has change in the vertical resolution. The vertical resolution of the water vapor is in the range 2.0 to 3.7 km from 316 to 0.22 hPa and along

track horizontal resolution varies from 210 to 360 km for pressure greater than 4.6 hPa. For
ozone, vertical resolution is ~2.5 km and the along track horizontal resolution varies between
300 and 450 km (Livesey et al., 2011). The Aura MLS gives around 3500 vertical profiles per
day and it crosses the equator at ~1:40 am and ~1:40 pm local time. For calculating the cross-
tropopause mass flux, we used ERA-Interim winds obtained during cyclone period.
We have taken the cyclone track information data from India Meteorological
Department (IMD) tropical cyclones observed best track data from year 2007-2013. During
this period, around 50 cyclones have formed over the north Indian Ocean. Due to the
considerable variability of cyclone life-cycles, for the present study we selected only 16
cyclones that lasted for more than 4 days. The tracks of all the cyclones used for the present
study are shown in Figure 1. Table 1 shows the classification of the cyclones over the North
Indian Ocean. The TCs over the north Indian ocean are classified in to different categories by
IMD based on their maximum sustained wind speed. There are classified as : (1) low pressure
when the maximum sustained wind speed at the sea surface is < 17 knots (32
km/hr),(2)depression (D) at 17–27 knots (32–50 km/hr), (3) deep depression (DD) at 28–33
knots (51–59 km/hr), (4) cyclonic storm (CS) at 34– 47 knots (60–90 km/hr), (5) severe
cyclonic storm (SCS) at 48–63 knots (90–110 km/hr), (6) very severe cyclonic storm (VSCS)
at 64–119 knots (119–220 km/hr), and (7) super cyclonic storm (SuCS) at > 119 knots (220
km/hr) (Pattnaik and Rama Rao, 2008). Table 2 shows the different cyclones used in the
present study and their maximum intensity, sustained time, and sustained time for peak
intensity period of the each cyclone. The mean sustained time for cyclones that occurred
during pre-monsoon, monsoon and post-monsoon seasons is 85.5 ± 52.4 hours, 122 ± 46.5
and 112.6 ± 29.47 hours, respectively. Out of the16 cyclones, 4 cyclones (CS-1, SCS-2and
VSCS-1)formed during pre-monsoon season, 3 cyclones formed during monsoon season (CS-
1, VSCS-1 and SuCS-1) and 9 cyclones (CS-1, SCS-2, and VSCS-6) formed during post-
monsoon season (Table 2).Depressions and deep depressions are not considered. The total
available MLS profiles for each cyclone that are used in the present study are listed in Table
2. We have 94 ± 21 mean MLS profiles for each cyclone used in the present study and when
segregated season wise, there are 108±6, 99±21 and 88±23 during monsoon, pre-monsoon
and post-monsoon season, respectively. The available total MLS profiles for each cyclone
vary with respect to sustained period of the cyclone and overall we have 1517 MLS profiles
within 1000 km from the cyclone centre from all the16 cyclones (Figure 2b). Since there are
(temporal) limitations in the satellite measurements, mean cross-tropopause flux is estimated
only for those cases of the cyclones that lasted for more than 4 days. However, our
quantification of the cross-tropopause flux will not be affected by this limitation as earlier
studies revealed that the maximum STE occurs during mature to peak stage of cyclone.
Details on the selection of 16 cyclones are presented in Ravindra Babu et al. (2015). In Figure
1, different colors indicate different categories of the cyclones.
**2.1. Tropopause characteristics observed during cyclones**

As mentioned earlier, in the tropical region the amount of water vapor transported into

the lower stratosphere from the troposphere is controlled by the cold tropical tropopause
temperatures (Fueglistaler et al., 2009).Large convection around the eye of the cyclone and
strong updrafts near the eye-walls transports large amount of water vapor into the lower
stratosphere through the tropopause. In this way, cyclones will affect the tropopause structure
(altitude/temperature). Thus, before quantification of STE, we show the tropopause
characteristics observed during the TCs. We used post-processed products of level 2 dry
temperature profiles with vertical resolution around 200 m provided by the COSMIC Data
Analysis and Archival Center (CDAAC) for estimating the tropopause parameters during
cyclones period from 2007-2013. COSMIC GPS-RO is a constellation of six microsatellites
equipped with GPS receivers (Anthes et al., 2008). We also used CHAllenging Minisatellite
Payload (CHAMP) GPS-RO data that are available between the years 2002 to 2006 and
COSMIC data from 2007-2013 for getting background climatology of tropopause parameters
over the north Indian Ocean.
Climatological mean of all the tropopause parameters are obtained by combining
GPS-RO measurements obtained from CHAMP and COSMIC (2002-2013). The tropopause
parameters include cold-point tropopause altitude (CPH) and temperature (CPT), lapse rate
tropopause altitude (LRH) and temperature (LRT) and the thickness of the tropical
tropopause layer (TTL), defined as the layer between convective outflow level (COH) and
CPH and are calculated for each profile of GPS-RO collected during the above mentioned
period. First, we separated the available RO profiles with respect to distance away from the
cyclone centre around 1000 km for individual cyclone for each day of the respective cyclone.
After separating, we calculated the tropopause parameters as mentioned above for each RO
profile. Total number of occultations used in the present study is shown in Figure 2(a). Then
we separated the tropopause parameters with respect to the different cyclone intensity. After
estimating the tropopause parameters for all the 16 TCs with respect to different intensity,
cyclone-centre composite of all tropopause parameters is obtained. After careful analysis, it is
found that there is no much variation in the tropopause parameters observed between D and
DD, and between CS and SCS, and thus they are combined to DD and CS, respectively. To
quantify the effect of the TCs on the tropopause characteristics, the climatological mean is
removed from the individual tropopause parameters. The climatological mean tropopause
parameters is estimated from the temperature profiles obtained by using GPS-RO data from
2002-2013. We also calculated the difference of tropopause parameters for different cyclone
intensities (Figures are not shown). Figure 3shows the cyclone centered – composite of mean
difference in the tropopause parameters (CPH, LRH, CPT, LRT, COH and TTL thickness)
between climatological mean (2002-2013) and individual tropopause parameters observed
during cyclones (irrespective of cyclone intensity) and the more detailed results on effect of
TCs on the tropopause variations and mean temperature structure in UTLS region during TCs
can be found in Ravindra Babu et al. (2015). We have reported that the CPH (LRH) is
lowered by 0.6 km (0.4 km) in most of the areas within the 500 km radius from the cyclone
centre and the temperature (CPT/LRT) is more or less colder or equal to the climatological
values from the area around 1000 km from the cyclone centre. Note that effect of cyclone can
be felt up to 2000 km but since the latitudinal variation also comes into picture when we
consider 2000 km radius, we restrict our discussion related to variability within 1000 km
from the cyclone centre. COH (TTL thickness) has increased (reduced) up to 2 km within 500
km from the cyclones and in some areas up to 1000 km. Note that this decrease in TTL
thickness is not only because of pushing up of the COH but also due to decrease of CPH.
From the above results, we concluded that the tropical tropopause is significantly affected by
the cyclones and the effect is more prominent within 500 km from the cyclone centre. These
changes in the tropopause parameters are expected to influence water vapor and ozone
transport in the UTLS region during cyclones.
**3. Results and discussion**
**3.1. Ozone variability in the UTLS region during cyclones**
To see the variability and the transport of ozone during the passage of cyclones, we
investigate the spatial and vertical variability of ozone in the UTLS region using MLS
satellite observations. As mentioned in Section 2.1, we also separated the MLS profiles based
on the distance from the TC centre for each day of the individual cyclone. From all the 16
cyclones cases, we separated the available MLS profiles with respect to distance from the
cyclone centre around 1000 km and also we separated the MLS profiles with respect to
different intensities of the cyclones. Figure 4shows the normalized cyclone centered –
composite of mean ozone mixing ratio (OMR) observed during cyclones (irrespective of
cyclone intensity) at 82hPa, 100hPa, 121hPa, and 146 hPa pressure levels during 2007-2013.
Note that we have reasonable number of MLS profiles (1517) from 16 cyclones to generate
the meaningful cyclone-centre composite of ozone. Black circles are drawn to show distances
250 km, 500 km, 750 km and 1000 km away from cyclone center. Since large variability in
OMR is noticed from one pressure level to other, we normalized the values to the highest
OMR value at a given pressure level. The highest OMR values at 82 hPa, 100 hPa, 121 hPa
and 146 hPa pressure levels is 0.38 ppmv, 0.28 ppmv, 0.19 ppmv and 0.13 ppmv,
respectively. Large spatial variations in the OMR are observed with respect to the cyclone
centre. At 82 hPa, higher OMR (~0.4 ppmv) in the South-West (SW) side up to 1000 km and
comparatively low OMR values (~0.2 ppmv) are noticed in the north of the cyclone centre.
At 100 hPa, an increase in the OMR (~0.2 ppmv) near the cyclone centre within 500 km is
clearly observed. This enhancement in OMR extends up to 146 hPa and is more prominent
slightly in the western and eastern side of the cyclone. In general, the large subsidence
located at the top of the cyclone centre is expected to bring lower stratospheric ozone to the
upper troposphere. This might be the reason for the enhancement of ozone in the cyclone
centre within 500 km. Earlier several studies have reported that the intrusion of the
stratospheric air in to the troposphere due to the subsidence in the eye region (Penn, 1965;
Baray et al., 1999; Das et al., 2009; Das et al., 2015). The present results also support this
aspect that the detrainment of ozone reached to the 146 hPa might be due to strong
subsidence. Interestingly, an enhancement in OMR in south east side at 121 hPa but not
either at 100 hPa or at 146 hPa can be noticed which need to be investigated further. Thus, in
general, higher ozone concentrations are observed in cyclone centre within 500 km and
slightly aligned to the western side of the cyclone centre.
In order to quantify the impact of cyclones on UTLS ozone more clearly we have
obtained anomalies by subtracting the mean cyclone-centered ozone observed during

cyclones from the background climatology of UTLS ozone that is calculated by using the total available MLS profiles from 2007-2013. Figure 4(e-h) shows the normalized mean difference of cyclone-centered ozone obtained after removing the background climatology values for different pressure levels shown in Figure 4(a-d). The maximum difference in OMR for corresponding normalized value at 82 hPa, 100 hPa, 121 hPa and 146 hPa pressure levels is -0.089 ppmv, -0.19 ppmv, -0.09 ppmv and -0.06 ppmv, respectively. Enhancement in the OMR (~0.1 ppmv) up to 1000 km from the cyclone centre is observed at 82 hPa. Interestingly, at 100 hPa OMR is more or less uniform throughout 1000 km from the cyclone centre except ~500 km radius from the centre where significant increase of OMR (~0.2 ppmv) is observed. This increase in the OMR is within 500 km from cyclone centre and extends up to 121 hPa. However, enhancement in OMR at 146 hPa extends up to 1000 km but distributed towards eastern and western sides of cyclone centre. Thus, it is clear that the detrainment of lower stratospheric ozone will reach up to 146 hPa during cyclone period due to presence of strong subsidence in the cyclone centre. We also calculated the cyclone-centre composite of ozone based on different cyclone intensities such as DD, SCS and VSCS. After carefully going through them, we have found that this detrainment of ozone reaching up to 146 hPa is more in the higher intensity period of the TCs. We do not know what happens below this pressure level due to limitation in the present data, however, studies (Das et al., 2015; Jiang et al., 2015) have shown that LS ozone can reach as low as boundary layer during cyclones. It will be interesting to see the variability in the water vapor as large amount of it is expected to cross the tropopause during the cyclone period and reach lower stratosphere.

**3.2. Water vapor variability in the UTLS region during cyclones**

As mentioned earlier, enormous amount of water vapor is expected to be pumped from lower troposphere to the upper troposphere and even it can penetrate into the lower stratosphere during cyclones. To see the linkage between tropopause variability and the

transport of water vapor during cyclones, we investigated the horizontal and vertical
variability of water vapor in the UTLS region using MLS satellite observations. Figure
5shows the normalized cyclone centered – composite of mean water vapor mixing ratio
observed during cyclones (irrespective of cyclone intensity) at 82hPa, 100hPa, 121hPa, and
146 hPa pressure levels observed by MLS during 2007-2013. Black circles are drawn to
shown the 250 km, 500 km, 750 km and 1000 km away from cyclone center. The highest
Water Vapor Mixing Ratio (WVMR)values for corresponding  normalized value at 82 hPa,
100 hPa, 121 hPa, and 146 hPa pressure levels is 4.44 ppmv, 4.49 ppmv, 6.9 ppmv and 16.03
ppmv, respectively. Significantly higher WVMR values are noticed extending from 500 km
up to 1000 km from the cyclone centre at 121 (~6.5 ppmv), 146 hPa (~15 ppmv) levels with
more prominence in the eastern side of the cyclon ecentre. Comparatively low values are
noticed in the centre of the cyclone, especially at 121 hPa. These results are comparing well
with higher WVMR observed in the eastern side of cyclones over Atlantic and Pacific Oceans
(Ray and Rosenlof, 2007). These results also compare well with those reported by Ravindra
Babu et al. (2015) where they used GPS-RO measured relative humidity and found
enhancement in RH in the eastern side of the centre in the upper troposphere (10-15 km) over
north Indian Ocean. The higher WVMR values are observed in the eastern side of the cyclone
centre might be due to the upper level anti-cyclonic circulation over the cyclones. It is
interesting to note that high WVMR lies not at the centre but extend from 500 to 1000 km
from the centre of cyclone. The WVMR show high at 121 and 146 hPa than at 100 and 82
hPa. It seems less water vapor has been transported to 100 and 82 hPa from below. As we
know, water vapor mostly origin from lower troposphere and decreasing with height. So
vertical transport of water vapor from the lower troposphere to the UTLS may lead to water
vapor enhanced at 121 and 146 hPa and some time it reaches to higher altitudes. The higher
WVMR presented at 100 and 82 hPa levels show the signature of the tropospheric air
entering even in to the lower stratosphere during cyclones.

In order to quantify the impact of cyclones on the UTLS water vapor more clearly, we

have obtained anomalies by subtracting the mean cyclone-centered water vapor observed
during cyclones from the background climatology mean of UTLS water vapor. Figure 5(e-h)
shows the normalized mean difference of the cyclone-centered WVMR obtained after
removing the background climatology values for different pressure levels shown in Figure
5(a-d). The maximum difference in WVMR for corresponding normalized values at 82 hPa,
100 hPa, 121 hPa, and 146 hPa pressure levels is -0.44 ppmv, -0.81 ppmv, -2.55 ppmv and -
9.09 ppmv, respectively. More than 7 ppmv differences are observed at 146 hPa within the
1000 km from the centre and at 121 hPa difference of ~ 2 ppmv is noticed extending up to
2000 km (figure not shown) in the eastern side of the centre. At 100 hPa and 82 hPa levels,
the increase in the WVMR is ~0.8 and ~0.6 ppmv, respectively, and the enhancement is more
observed in the NE side of the cyclone centre. Thus, a clear STE is evident during the
cyclone over north Indian Ocean where a clear enhancement in the water vapor (ozone) in the
lower stratosphere (upper troposphere) is observed. For quantifying the amount of STE, we
calculated the cross-tropopause mass flux for each cyclone by considering the spatial extent
within the 500 km from the cyclone centre and results are presented in the following sub-
section.
**3.3. Cross tropopause flux observed during cyclones**

We adopted method given by Wei (1987) to estimate the cross tropopause mass flux,

*F*. *F* is defined as:
$F = \frac{1}{g}\left(-\omega + V_h . \nabla P_{tp} + \frac{\partial P_{tp}}{\partial t}\right) = \left(-\frac{\omega}{g} + \frac{1}{g}V_h . \nabla P_{tp}\right) + \frac{1}{g}\frac{\partial P_{tp}}{\partial t} = F_{AM} + F_{TM}$        (1)
where ω is the vertical pressure-velocity, $V_h$ is the horizontal vector wind, $P_{tp}$ is the pressure
at the tropopause, g is the acceleration due to gravity, $F_{AM}$ is the air mass exchange due to
horizontal and vertical air motions, $F_{TM}$ is the air mass exchange due to tropopause motion.

The wind information is taken from ERA-Interim, and the tropopause temperature and

pressure within 500 km from the cyclone centre is estimated from COSMIC GPS-RO
measurements (Ravindra Babu et al., 2015). These values are considered for the maximum
intensity day for each of the 16 cyclones and the respective cross tropopause flux is
estimated. Since the above mentioned results showed that the higher OMR values are
observed in the west and NW side and more water vapor is located at the eastern side of the
cyclone centre, we separated the area into 4 sectors with respect to cyclone centre as C1 (NW
side), C2 (NE side), C3 (SW side), and C4 (SE side), respectively as shown in Figure 4(a).
List of cyclones used in the present study with their names, cyclone intensity (CI), centre
latitude, centre longitude, minimum estimated central pressure on their peak intensify day are
provided in Table 3. The total flux $F$ (equation 1) depends on the air mass exchange due to
horizontal and vertical air motion ($F_{AM}$), and the air mass exchange due to tropopause motion
itself ($F_{TM}$). Since number of COSMIC GPS-RO measurements are not sufficient to estimate
the second term ($F_{TM}$) for each event, we calculated only the first part of the equation ($F_{AM}$)
individually for each of cyclone with respect to different sectors mentioned above and the
values are presented in Table 3.However, we roughly estimated the contribution of second
term by assuming change in the tropopause pressure by 0.5 hPa increase (decrease) within 6
hr and could see cross-tropopause flux for CS is $0.25\pm0.07 \times 10^{-3}$ kgm$^{-2}$s$^{-1}$ ($-0.36\pm0.07\times10^{-3}$
kgm$^{-2}$s$^{-1}$) and for VSCS it is $-0.24\pm0.3\times10^{-3}$ kgm$^{-2}$s$^{-1}$ ($-0.85\pm0.3\times10^{-3}$ kgm$^{-2}$s$^{-1}$). If there is
change in the tropopause pressure by 1 hPa increase (decrease), the flux for CS is
$0.55\pm0.07\times10^{-3}$ kgm$^{-2}$s$^{-1}$ ($-0.66\pm0.07\times10^{-3}$ kgm$^{-2}$s$^{-1}$) and for VSCS it is $0.06\pm0.3\times10^{-3}$ kgm$^{-2}$s$^{-1}$
($-1.16\pm0.3\times10^{-3}$ kgm$^{-2}$s$^{-1}$).
Figure 6 shows the cross-tropopause flux estimated in each sector from the centre of
the cyclone for the different cyclone intensities (estimated based on the cyclone centre
pressure). Red lines show the best fit. It clearly shows that the downward flux is always more
in C1 and C3 sectors, whereas C2 sector show more upward flux. The flux itself varies with
the cyclone intensity and it is found that the increase in downward flux as the cyclone centre
pressure decreases particularly forC1 and C3 sectors. Whereas, in C4 sector, increase in the
upward flux is seen as the cyclone intensity increases but always upward in C2 sector,
irrespective of the cyclone intensity. The second term (in equation 1) itself corresponds the
air mass exchange from the tropopause motion and generally during cyclone period there is
an ~400 m difference in tropopause altitude (LRH) within 500 km from the centre of the
cyclone (Figure 3).Thus, the spatial and temporal variation of the tropopause during the
cyclones itself is very important for to decide the flux as downward or upward. Interestingly,
C1 and C3 sectors of cyclone show dominant downward mean flux and C2 and C4 sectors
show dominant upward mean flux with the values of $0.4\pm0.4\times10^{-3}$kgm$^{-2}$, $1.2\pm1.0\times10^{-3}$kgm$^{-2}$,
$0.2\pm0.1\times10^{-3}$ kgm$^{-2}$and $0.12\pm0.3\times10^{-3}$ kgm$^{-2}$, respectively. These results strongly support our
findings of higher ozone in the NW and SW sides and higher water vapor in the NE side of
the cyclone centre. The mean flux is observed to vary with the intensity of the cyclone.  Mean
flux for the severe cyclonic storms (CS) is $-0.05\pm0.29\times10^{-3}$ kgm$^{-2}$whereas for very severe
cyclonic storms (VSCS) it is $-0.5\pm1.07\times10^{-3}$kgm$^{-2}$. Reutter et al. (2015) reported the upward
and downward mass fluxes across the tropopause are more dominant in a deeper cyclones
compared to a less intense cyclones over the North Atlantic. Our results are comparable with
their results with the averaged mass flux of the stratosphere to troposphere as $0.3\times10^{-3}$ kgm$^{-2}$
s$^{-1}$ (340 kgkm$^{-2}$ s$^{-1}$) in the vicinity of cyclones over the North Atlantic Ocean. They also
reported that the more transport across the tropopause occurred in the west side of the
cyclone centre during intensifying and mature stages of the cyclones over the North Atlantic
region.
**4. Summary and conclusions**
In this study, we have investigated the vertical and spatial variability of ozone and
water vapor in the UTLS region during the passage of cyclones occurred between 2007 and
2013 over the North Indian Ocean by using Aura-MLS satellite observations. In order to
make quantitative estimate of the impact of cyclones on the ozone and water vapor budget in
the UTLS region, we removed the mean cyclone-centre ozone and water vapor from the
climatological mean calculated using MLS data from 2007 to 2013. We estimated the mean
cross- tropopause flux for each of the cyclones on their peak intensity day. The main findings
are summarized below.
1. Lowering of the CPH (0.6 km) and LRH (0.4 km) values with the coldest CPT and
LRT (2–3 K) within a 500 km radius from the cyclone centre is noticed. Higher (2
km) COH leading to the lowering of TTL thickness (~3 km) is clearly observed
(Ravindra Babu et al., 2015).
2. The impact of cyclones on ozone and the tropopause (altitude/temperature) is more
prominent within 500 km from the cyclone centre, whereas it is high from 500 km to
1000km in case of water vapor.
3. Detrainment of ozone is highest in the cyclone centre (within 500 km from the centre)
due to strong subsidence over top of the cyclone centre and this detrained ozone
reaches as low as 146 hPa level (~13-14 km).
4. The detrainment of ozone is more in the higher intensity period (SCS or VSCS) of the
cyclone compared to the low intensity (D or DD).
5. Interestingly, significant enhancement in the lower stratospheric (82 hPa) water vapor
is noticed in the east and southeast side from the cyclone centre.
6. Dominant downward [upward] cross-tropopause flux is observed in C1 (NW) and C3

(SW) [C2 (NE) and C4 (SE)] sectors of the cyclone.

Figure 7 shows the typical structure (not to scale) of the TC along with convective towers,
updrafts, downdrafts which above mentioned tropopause variability with respect to cyclone
centre  in the form of the schematic diagram. This figure is re-drawn from the basic idea
given in Chapter 9 and figure 6 of www.geology.sdsu.edu. The results presented in Figure 4
and Figure 5 is a composite picture of all 16 cyclones. Because each individual cyclone is
different, this composite picture will differ somewhat from the typical structure shown in
Figure 7. The tropopause altitude (CPH) is lowered by 0.6 km within 500 km from the centre
of the cyclone. The convective out flow level (COH) slightly pushes up (~2 km) with in 500
km from the centre of the cyclone but not exactly in the centre. Thus, a decrease of about 3
km in the TTL thickness is observed within the 500 km from the cyclone centre. Cyclone
includes eye that extends from few km to 10's of kilometers. Strong convective towers with
strong updrafts extending up to the tropopause in the form of spiral bands extending from 500
to 1000 km are present. Strong water vapor transport in to the lower stratosphere (82 hPa)
while pushing up the COH is observed around these spiral bands in the present study.
Between these spiral bands equal amount of subsidence is expected with strong subsidence
existing at the centre of the cyclone. Significant detrainment of ozone present above or
advected from the surroundings is observed reaching as low as 146 hPa at the cyclones
centre. Thus, it is clear that ozone reaches upper troposphere from lower stratosphere through
the centre of the cyclone, whereas water vapor transport in to the lower stratosphere will
happen from the 500 to 1000 km from the cyclones centre. Since  more intense cyclones are
expected to occur in a changing climate (Kuntson et al., 2010), the amount of water vapor
and ozone reaching to the lower stratosphere and upper troposphere, respectively, is expected
to increase thus affecting complete tropospheric weather and climate. Future studies should
focus on these trends.
**Acknowledgements:** We would like to thank COSMIC Data Analysis and Archive Centre
(CDAAC) for providing GPS-RO data used in the present study through their FTP site
(http://cdaac-www.cosmic.ucar.edu/cdaac/products.html). The provision of tropical cyclone
best track data used in the present study by IMD through their website
(http://www.imd.gov.in/section/nhac/dynamic/cyclone.htm) and Aura-MLS observations
obtained from the GES DISC through their ftp site (https://mls.jpl.nasa.gov/index-eos-
mls.php) is highly acknowledged. This work is supported by Indian Space Research
Organization (ISRO) through CAWSES India Phase-II Theme 3 programme. The authors
would like to thank the Editor Dr. Rolf Müller, and two anonymous reviewers whose
comments helped considerably in improving the quality of this paper

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

**Figure captions:**
**Figure 1.** Tropical cyclone tracks of different categories (cyclonic storm (CS, blue color),
severe cyclonic storm (SCS, orange color), very severe cyclonic storm (VSCS, red color)
and super cyclonic storm (SuCs, magenta color)) that occurred over North Indian Ocean
during 2007 - 2013.
**Figure 2.** Cyclone-centred composite of total available (a) COSMIC GPS RO occultations
and (b) MLS profiles obtained from all the 16 cyclones that are used in the present study.
**Figure 3.** Cyclone centered – composite of mean difference in the tropopause parameters
between climatological mean (2002-2013) and individual tropopause parameters observed
during cyclones(irrespective of cyclone intensity) in (a) CPH (km), (b) LRH (km), (c) CPT
(K), (d) LRT (K), (e) COH (km) and (f) TTL thickness (km). Black circles are drawn to
show the 250 km, 500 km, 750 km and 1000 km away from cyclone center.
**Figure 4.** Normalized cyclone centered – composite of mean ozone mixing ratio observed
during cyclones (irrespective of cyclone intensity) at (a) 82hPa, (b) 100hPa, (c) 121hPa, (d)
146 hPa levels by MLS during 2007-2013. (e) to (h) same as (a) to (d) but for normalized
mean difference in the ozone mixing ratio between climatological mean (2007-2013) and
individual events.  Black circles are drawn to show the 250 km, 500 km, 750 km and 1000
km away from cyclone center. Sectors showing C1 (NW), C2 (NE), C3 (SW) and C4 (SE)
are also shown in (a).
**Figure 5.** Same as Fig. 4, but for water vapor mixing ratio.
**Figure 6.** Cross-tropopause flux estimated in the (a) C1 (NW), (b) C2 (NE), (c) C3 (SW), and
(d) C4 (SE) sectors from the centre of cyclone for different cyclone intensities (estimated
based on cyclone centre pressure). Red lines show the best fit.
**Figure 7.** Schematic diagram showing the variability of CPH (brown color line) and COH
(magenta color line) with respect to the centre of cyclone. Spiral bands of convective

towers reaching as high as COH are shown with blue color lines. Light blue (red) color up (down) side arrow shows the up drafts (downdrafts/subsidence). Thickness of the arrows indicates the intensity.

**Table captions:**

**Table1**.Classification of cyclonic systems over the north Indian Ocean.

**Table 2.** Tropical cyclones occurred during different seasons, cyclone name, cyclone Intensity (CI), cyclone period, total sustained time, Sustained time with maximum intensity and total number of available MLS profiles

**Table 3.** Cyclone name, cyclone Intensity (CI), centre latitude, centre longitude, estimated central pressure and estimated cross-tropopause mass flux with respect to cyclone centre for C1 (NW side), C2 (NE side), C3 (SW side) and C4 (SE side), respectively.

**Tables:**
**Table1.**IMD classification of cyclonic systems over the north Indian Ocean.

| Intensity of the system | Maximum sustained surface winds (knots) at sea (1 knot =0.5144 m/s) |
|---|---|
| Low pressure area | <17 |
| Depression | 17–27 |
| Deep depression (DD) | 28–33 |
| Cyclonic storm (CS) | 34-47 |
| Severe cyclonic storm (SCS) | 48-63 |
| Very severe cyclonic storm (VSCS) | 64–119 |
| Super cyclonic storm (SuCS) | >119 |













**Table 2.** Tropical cyclones occurred during different seasons, cyclone name, cyclone
Intensity (CI), cyclone period, total sustained time, Sustained time with maximum intensity
and total number of available MLS profiles.

| Season | Cyclone Name | Cyclone Intensity (CI) | Cyclone Period (days) | Total Sustained time (hours) | Sustained Time with maximum intensity (hours) | Total available MLS profiles |
|---|---|---|---|---|---|---|
| Monsoon (JJA) | 03B(2007) | CS | >4 | 75 | 6 | 104 |
| | PHET (2010) | VSCS | >4 | 168 | 42 | 116 |
| | Gonu (2007) | ScCS | >4 | 123 | 72 | 105 |
| Pre-Monsoon (MAM) | Mahasen(2013) | CS | >4 | 24 | 24 | 119 |
| | Aila (2009) | SCS | 4 | 72 | 9 | 79 |
| | Laila (2010) | SCS | 4 | 96 | 27 | 82 |
| | Nargis (2008) | VSCS | >4 | 150 | 87 | 118 |
| Post-Monsoon (SON) | Nilam (2012) | CS | >4 | 102 | 36 | 52 |
| | Jal (2010) | SCS | 4 | 99 | 30 | 75 |
| | Helen (2013) | SCS | 4 | 78 | 30 | 72 |
| | Giri (2010) | VSCS | 4 | 66 | 15 | 65 |
| | Phailin (2013) | VSCS | >4 | 147 | 66 | 111 |
| | Leher (2013) | VSCS | >4 | 114 | 36 | 111 |
| | SIDR (2007) | VSCS | >4 | 138 | 72 | 114 |
| Winter (DJF) | Madi (2013) | VSCS | >4 | 150 | 36 | 104 |
| | Thane (2011) | VSCS | >4 | 120 | 36 | 90 |




**Table 3.** Cyclone name, cyclone Intensity (CI), centre latitude, centre longitude, estimated
central pressure and estimated cross-tropopause mass flux with respect to cyclone centre
for C1 (NW side), C2 (NE side), C3 (SW side) and C4 (SE side), respectively.

| | | | | | Flux @500km | | | |
|---|---|---|---|---|---|---|---|---|
| Cyclone | CI | Centre Latitude | Centre Longitude | Estimated Central Pressure (hPa) | C1 | C2 | C3 | C4 |
| 03B | CS | 23.5 | 66 | 986 (25Jun2007) | -0.013 | 0.661 | -0.603 | -0.258 |
| Aila | SCS | 22 | 88 | 968 (25May2009) | 1.90E-04 | 0.191 | -0.299 | -0.072 |
| Helen | SCS | 16.1 | 82.7 | 990 (21Nov2013) | 0.025 | 0.216 | -0.095 | -0.11 |
| Jal | SCS | 11 | 84 | 988(6Nov2010) | 0.025 | 0.384 | -0.4 | -0.218 |
| Laila | SCS | 14.5 | 81 | 986 (19May2010) | -0.012 | 0.123 | -0.352 | -0.299 |
| Mahasen | CS | 18.5 | 88.5 | 990 (15May2013) | -0.006 | 0.354 | -0.473 | -0.256 |
| Nilam | CS | 11.5 | 81 | 990 (31Oct2012) | 0.016 | 0.313 | -0.274 | -0.097 |
| Nargis | VSCS | 16 | 94 | 962 (2May2008) | -0.828 | 0.094 | -1.946 | 0.384 |
| Giri | VSCS | 19.8 | 93.5 | 950 (22Oct2010) | -0.518 | 0.022 | -0.823 | 0.032 |
| Gonu | SuCS | 20 | 64 | 920 (4Jun2007) | -0.502 | 0.123 | -2.563 | 0.37 |
| Lehar | VSCS | 13.2 | 87.5 | 980 (26Nov2013) | -0.55 | 0.119 | -2.019 | 0.411 |
| Madi | VSCS | 13.4 | 84.7 | 986 (10Dec2013) | -0.375 | 0.054 | -1.449 | 0.352 |
| Phailin | VSCS | 18.1 | 85.7 | 940 (11Oct2013) | -0.9 | 0.179 | -2.576 | 0.479 |
| Phet | VSCS | 18 | 60.5 | 964 (2Jun2010) | -1.058 | 0.203 | -2.698 | 0.559 |
| SIDR | VSCS | 19.5 | 89 | 944 (15Nov2007) | -0.493 | 0.066 | -0.926 | 0.231 |
| Thane | VSCS | 11.8 | 80.6 | 970 (29Dec2011) | -1.272 | 0.356 | -2.979 | 0.558 |



**Figures:**

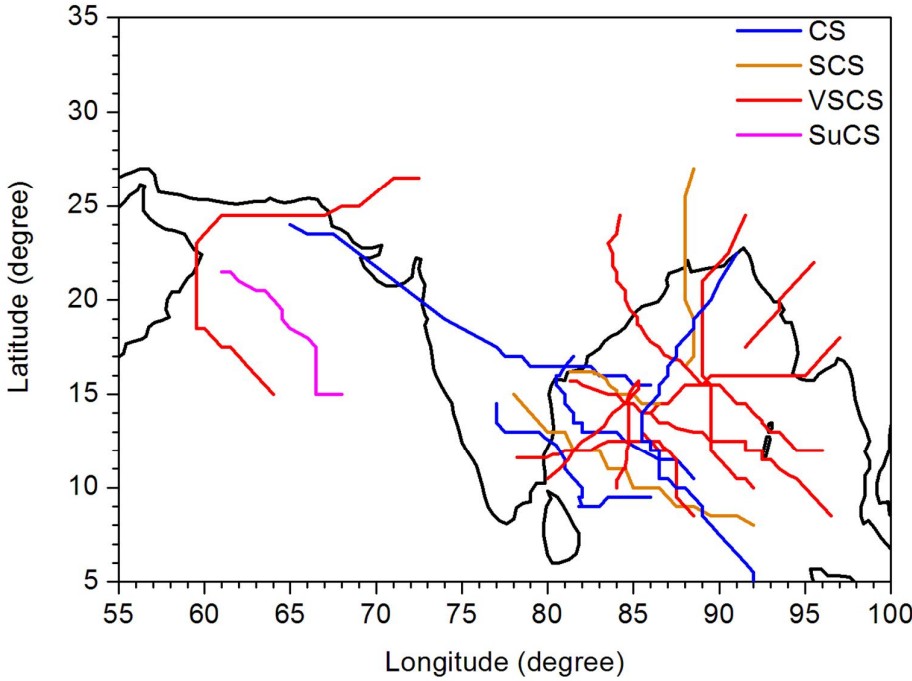


**Figure 1.** Tropical cyclone tracks of different categories (cyclonic storm (CS, blue color),

severe cyclonic storm (SCS, orange color), very severe cyclonic storm (VSCS, red color)

and super cyclonic storm (SuCs, magenta color)) that occurred over North Indian Ocean

during 2007 - 2013.



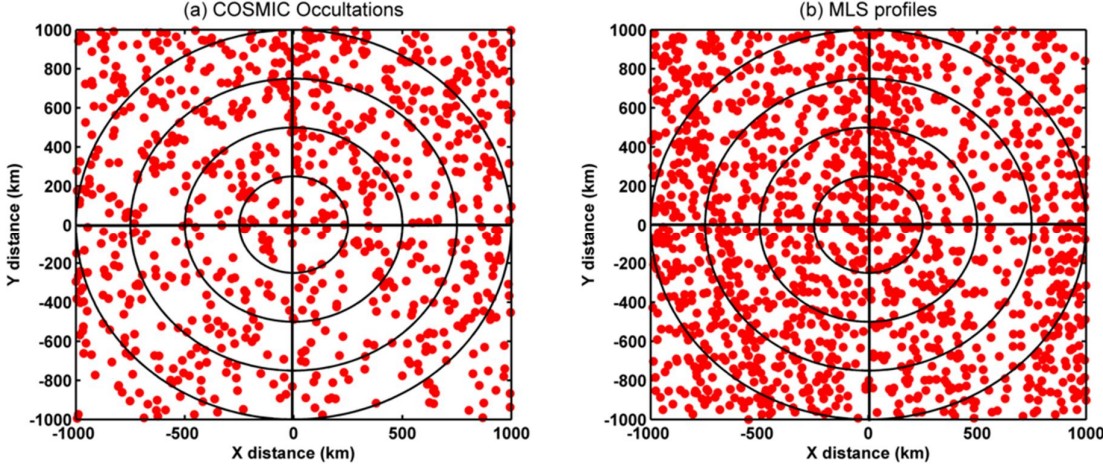


**Figure 2.** Cyclone-centred composite of total available (a) COSMIC GPS RO occultations

and (b) MLS profiles obtained from all the 16 cyclones that are used in the present study.

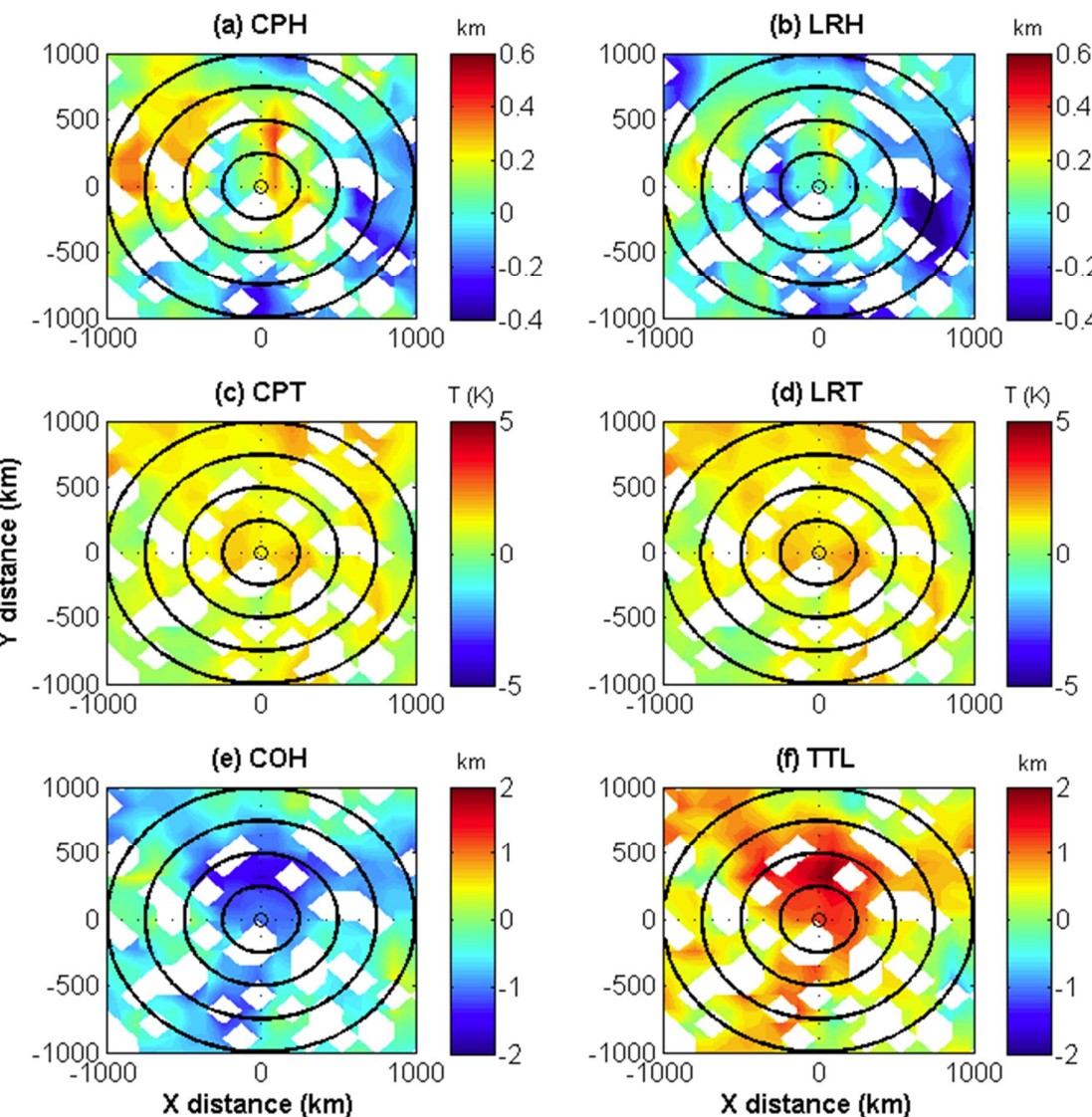


**Figure 3.** Cyclone centered – composite of mean difference in the tropopause parameters between climatological mean (2002-2013) and individual tropopause parameters observed during cyclones (irrespective of cyclone intensity) in (a) CPH (km), (b) LRH (km), (c) CPT (K), (d) LRT (K), (e) COH (km) and (f) TTL thickness (km). Black circles are drawn to show the 250 km, 500 km, 750 km and 1000 km away from cyclone center (taken from Ravindra Babu et al., ACP, 2015).

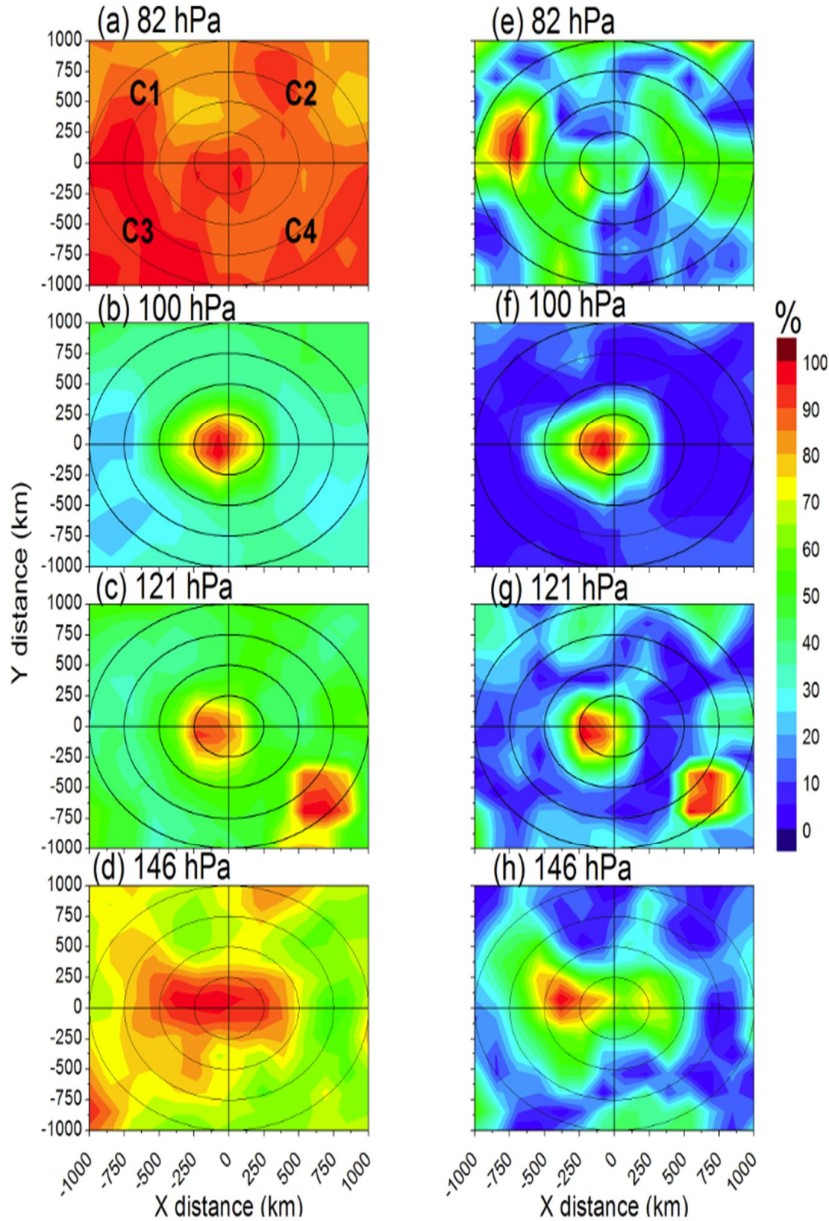

**Figure 4.** Normalized cyclone centered – composite of mean ozone mixing ratio observed

during cyclones (irrespective of cyclone intensity) at (a) 82hPa, (b) 100hPa, (c) 121hPa,

(d) 146 hPa levels by MLS during 2007-2013. (e) to (h) same as (a) to (d) but for

normalized mean difference in the ozone mixing ratio between climatological mean (2007-

2013) and individual events. Black circles are drawn to show the 250 km, 500 km, 750 km

and 1000 km away from cyclone center. Sectors showing C1 (NW), C2 (NE), C3 (SW)

and C4 (SE) are also shown in (a).

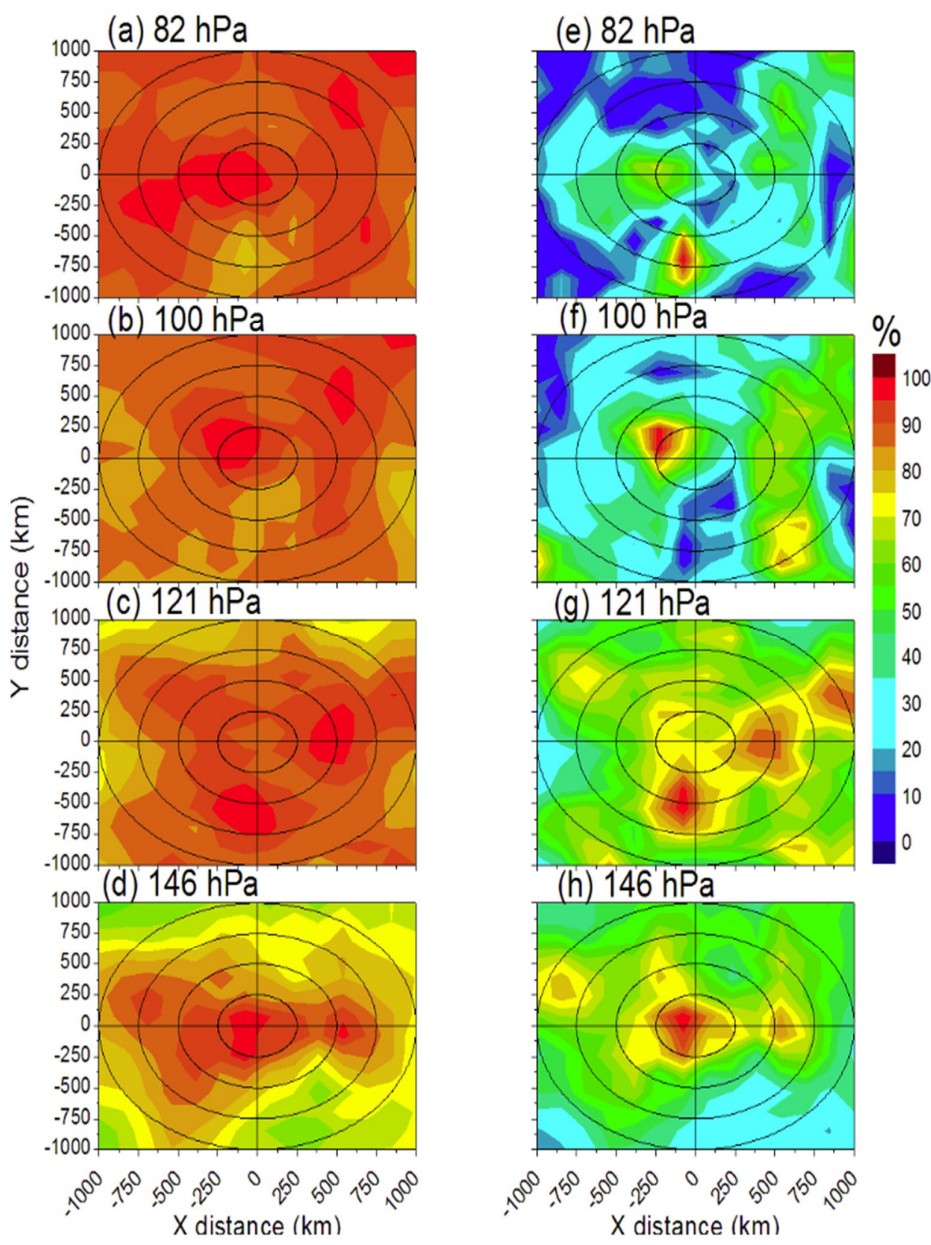


**Figure 5.** Same as Fig. 4, but for water vapor mixing ratio.



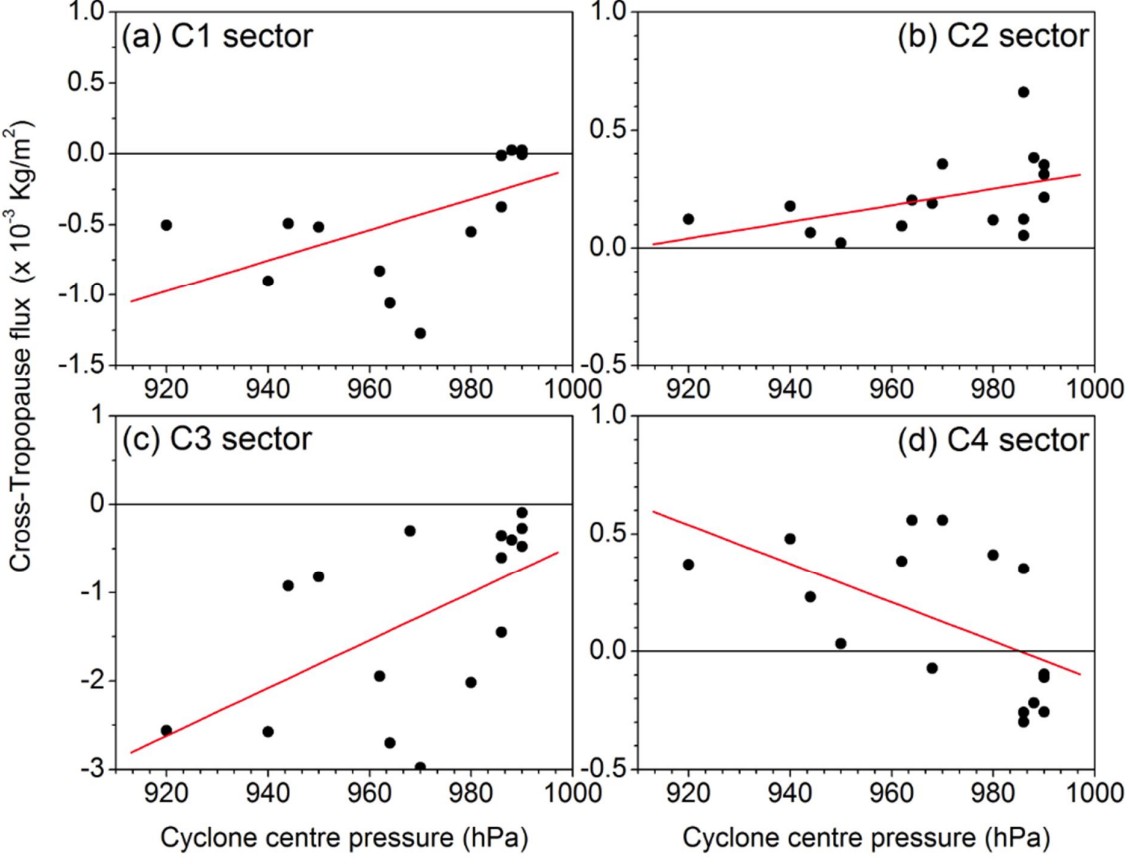


**Figure 6.** Cross-tropopause flux estimated in the (a) C1 (NW), (b) C2 (NE), (c) C3 (SW), and

(d) C4 (SE) sectors from the centre of cyclone for different cyclone intensities (estimated

based on cyclone centre pressure). Red lines show the best fit.






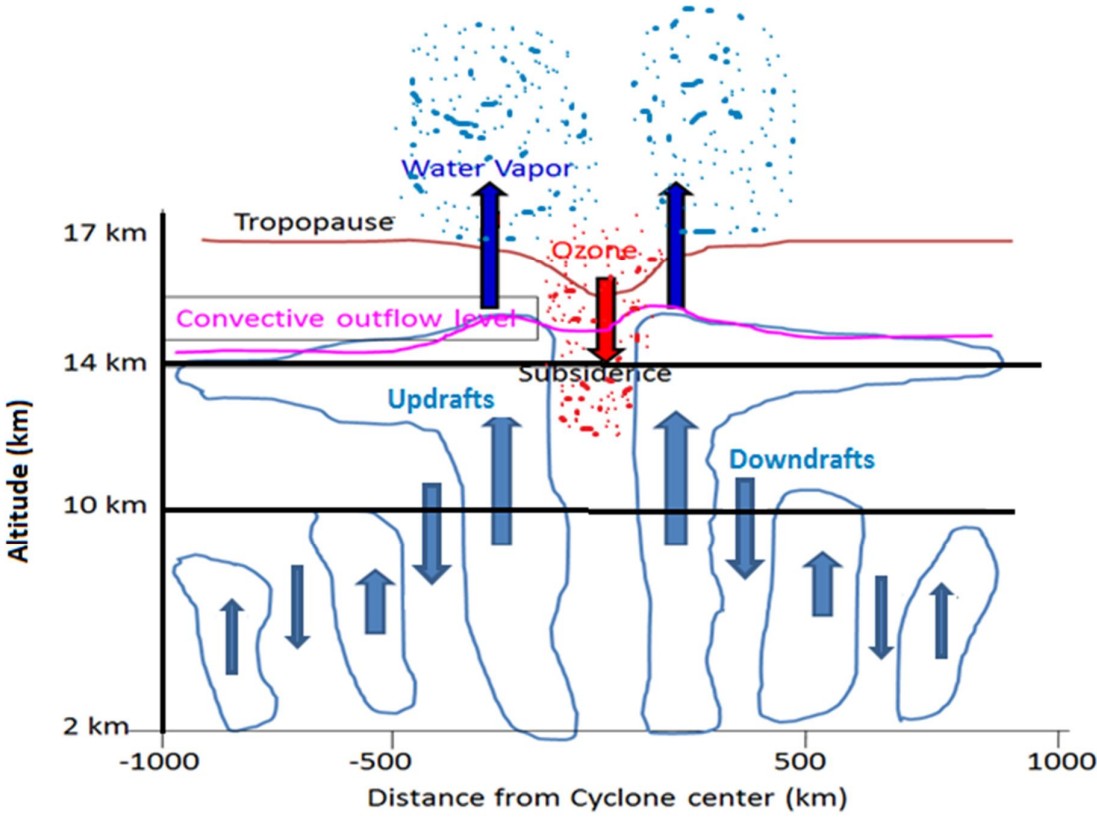


**Figure 7.** Schematic diagram showing the variability of CPH (brown color line) and COH

(magenta color line) with respect to the centre of cyclone. Spiral bands of convective

towers reaching as high as COH are shown with blue color lines. Light blue (red) color up

(down) side arrow shows the up drafts (downdrafts/subsidence). Thickness of the arrows

indicates the intensity. This figure is re-drawn from the basic idea given in figure 6 of

www.geology.sdsu.edu/visualgeology/naturaldisasters/Chapters/Chapter9Cyclones.pdf .
