# Peer review of "1. Introduction"

_Atmospheric Chemistry and Physics, 2015_

## Referee Comment (RC1) · Anonymous Referee #2 · 29 Feb 2016

[12pt]article

mhchem

**Reviewer Comment**

The paper presents the impact of cyclones that occurred over the North Indian Ocean during 2007-2013 on stratosphere-troposphere exchange using satellite measurements. Changes in ozone and water vapor distribution in the upper troposphere and lower stratosphere were analyzed. The cross-tropopause mass flux was estimated. The manuscript has some significant shortcomings. Therefore, I recommend some important revisions to address the comments listed below before publication by ACP.

**General comments:**

1) Scientific significance
The paper presents new interesting results, however the results need to be better developed.

2) Scientific quality
One important questions is whether the MLS measurements have sufficient spatial and temporal resolution to apply the used methodology? This has to be demonstrated. The explanation how the cross tropopause mass flux is calculated and which data are used is confusing. The method is explained in Sect. 2 and the used data are introduced in Sect. 3.1. I recommend to combine this in one Section.
Further, the method of Ravindra Babu et al., 2015 is used (e.g. Fig. 2). However, the reader can not understand this method without reading Babu et al., 2015. I recommend to provide more information about this method in Sect. 2.
Many general statements have not been established with references (e.g. within the introduction, see below specific comments).

3) Presentation quality
The presentation quality needs some improvements. There are number of language and grammar issues. Further a lot of blank characters are missing, in particular after mathematical symbols or brackets. In the manuscript, abbreviations are still used that

are not introduced. In some figures, the legend is missing.

**Specific comments:**

1. Introduction:

p. 3, line 51: 'Tropical cyclones with deep convective synoptic scale systems persisting for a few days to weeks play an important role on the mass exchange between troposphere and stratosphere and vice versa.'
Please add some references.

p. 3, line 52: 'They transport large amount of water vapor, energy and momentum to the upper troposphere and lower stratosphere (UTLS) region.'
Please add some references.

p. 3, line 60: 'The transport of water vapour and ozone around the tropopause caused by the cyclones can affect the radiation balance of the atmosphere.'
Please add some references.'

p. 3, line 62: 'Increase of water vapor in the LS region will leads to a warming and ozone loss in this atmospheric region (Stenke and Grewe, 2005).'

An increase of stratospheric water vapor contributes to tropospheric warming and stratospheric cooling, see e.g.:

Climate Change 2007: The Physical Science Basis. Contribution of Working Group I

**[ACPD](...)**
to the Fourth Assessment Report of the Intergovernmental Panel on Climate Change, ed. S. Solomon, D. Qin, M. Manning, M. Marquis, K. Averyt, M. M. B. Tignor, H. L. Miller and Z. Chen, Cambridge University Press, Cambridge, UK, and New York, NY, USA, 2007, pp. 1-996.

D. Rind and P. Lonergan, J. Geophys. Res., 1995, 100, 7381-7396

P. Forster and K. P. Shine, Geophys. Res. Lett., 1999, 26, 3309-3312

V. L. Dvortsov and S. Solomon, J. Geophys. Res., 2001, 106, 7505-7514.

D. T. Shindell, Geophys. Res. Lett., 2001, 28, 1551-1554.

P. Forster and K. P. Shine, Geophys. Res. Lett., 2002, 29, 1086-1089.

G. Myhre, S. J. Nilsen, L. Gulstad, K. P. Shine, B. Rognerud and I. S. A. Isaksen, Geophys. Res. Lett., 2007, 34, L01807.

However, small changes of water vapor in the lower stratosphere have an impact on surface climate, see e.g:

Riese et al., Impact of uncertainties in atmospheric mixing on simulated UTLS composition and related radiative effects, J. Geophys. Res., 117, D16305, doi:10.1029/2012JD017751, 2012.

Solomon et al., Contributions of stratospheric water vapor to decadal changes in the rate of global warming, Science, 327, 1219-1223, 2010.

p. 3, line 65: troposphere air → tropospheric air

p. 4, line 82: 'TC event': abbreviation is not explained

p. 4, line 86: 'MST Radar observations': abbreviation is not explained

p. 4, line 87: 'BoB': abbreviation is not explained

p. 4, line 87: 'More literature related to influence of cyclones on the UTLS structure and composition is presented in Cairo et al. (2008).'
Unspecific statement: please add some details or remove Cairo et al. 2008.

p. 5, line 105: 'COSMIC' is not explained

2. Data and Methodology

p. 5, line 116: How many MLS profiles or measurements (spatial and temporal resolution, horizontal distance between tracks) contribute to one typhoon event. Please add here some information and demonstrate that the data density is sufficient.

p. 5, line 120: Which definition is used for the tropopause?

[Figure]

p. 6, line 135: Please add the precise time period for pre- and post-monsoon season and explain why you exclude the monsoon season.

p. 7, line 149: 'tropopause parameters': Which parameters? Please combine this paragraph with details from Sect. 3.1'.

3. Results and discussion

p.7, line 162: How are the climatological mean values calculated? Is the monsoon season in the climatological mean excluded? During the Asian monsoon season the tropopause above the Asian monsoon anticyclone is elevated. Therefore, during this time period the lapse rate tropopause altitude differs from the altitude during the rest of the year. Is this considered in your analysis?

p. 7, line 169: How many measurements (tracks) do you have within 1000 km radius for one cyclone?

p. 7, line 175: How is the cyclone intensity considered in the methodology of Ravindra Babu et. al, 2015? Please give a short summary about the method of Ravindra Babu et al., 2015 used for Fig. 2. How is vertical uplift at different flanks of the cyclone and difference between individual cyclones considered?

p. 12, line 280-284: '...higher ozone mixing ratios are observed in the western and northwest side and more water vapor is located at the eastern side of the cyclonic center....' Why do you have this preference for the western and eastern side, respectively? In the schematic diagram Fig. 6 upward and downward transport of water vapor and

ozone is shown. The diagram implies rotational symmetry around the center of the cyclone. How fits the rotational symmetry together with the preference at the western and eastern side?

p. 12, line 294: 'by assuming change in the tropopause pressure by 0.5 hPa' Why 0.5 hPa is used?

p. 13, line 299: Please explain why different cross-tropopause flux occurs in different sectors.

4. Summary and conclusions

p. 14, line 335: 'The main findings of the present communication are summarized below.' → Our main findings are summarized below.'

p. 14, line 336-339: 'Lowering of CPH (0.6 km) and LRH (0.4 km) values with coldest CPT and LRT (2-3K) within a 500 km radius from the cyclone centre is noticed. Higher (2 km) COH leading to the lowering of TTL thickness ( 3 km) is clearly observed (Ravindra Babu et al., 2015).' That is a result from Ravindra Babu et al, 2015 and not from the present paper Ratman et al.. That should be clearly recognizable in the text.

p. 15, line 346-347: 'Interestingly significant enhancement in the lower stratosphere (82 hPa) water vapor is noticed in the east and SE side from the cyclone centre.' Again, why only at the east and SE side?

p. 15, line 355-357: 'Strong convective towers with strong updrafts extending up to

the tropopause altitude in the form of spiral bands extending from 500 to 1000 km are present.' In Fig. 6, three bands of downward transport of ozone and three bands for upward transport of water vapor are drawn which are not visible in Fig. 3 and 4. Please explain this discrepancy or adapt Fig. 6. To confirm the spiral bands of upward and downward transport illustrated in Fig. 6 trajectory calculations would be very helpful.

Figures:

Fig. 1: 'strom' → 'storm'

Fig. 3: Legend from a-d is missing.

Fig. 4: Legend from a-d is missing.

---

## Referee Comment (RC2) · Anonymous Referee #3 · 16 Mar 2016

This is an interesting study of the impact of cyclones on ozone and water vapour in the upper troposphere and lower stratosphere. It is based on the analysis of some cases study, using satellite measurements to estimate the air flux across the tropopause. It is surely a valuable contribution on a hot topic in stratospheric research, since the ability to predict future changes in the stratosphere relies on correct estimates on how tropical troposphere to stratosphere transport might evolve. I definitely agree that the role of deep convection in cyclones is worth of more research, and it is appropriate for the journal, so I encourage the publication of this work. However, there are a number of open issues that have to be addressed, therefore I recommend a revision before publication. I had the chance to read the general comments of the Anonymous Reviewer #2 and I do share all his/her general comments. In particular I find strange how the results from previous work of Ravindra Babu et al., 2015 are used in the present paper: on one hand, figures and conclusions from that paper are reproduced in a way that seems redundant, on the other hand a description of the method used in that work, which is duplicated in the present one, is lacking so to force the reader to go to the original reference. I therefore suggest to briefly summarize the results AND methods presented in Ravindra Babu et al., and to skip fig.2. Detailed comments: lines 49-52: These sentences seems more to describe what the article is aimed for, than an introduction, The authors should support their claims with references, or the sentences should be made less assertive. 61: Again, the assessment of the effectiveness of cyclones in promoting STE is the objective of the paper. References should be made to previous studies supporting this claim, or the sentence should be dropped, or reformulated to introduce the aim of the paper. 62-63: The Stenke and Grewe paper deals mainly with the impact of water vapour increase on ozone chemistry. I did not find any claim of temperature increase induced by an increase of WV, there. On the contrary, there is a lot of modeling evidence (and even some experimental study, see as instance Maycock et al., Q. J. R. Meteorol. Soc., 2014), in the literature, that an increase stratospheric water vapor would lead to a cooling of stratospheric temperatures. So the sentence in the paper seems not correct. 82, 86, 87,96: TC, MST, BoB, COSMIC, abbreviations have not been introduced earlier. 91: The findings presented in Cairo et al. (2008) should be reported. 128-134: Such information should be presented as a table. 139-140: This sentences is not clear. Is it suggesting that only long lasting cyclones have been selected in order to have enough MLS WV profiles in the cyclone area? This is quite an important point, and the average number of MLS profiles used should be quoted, maybe even in the form of a table, for each cyclone (the developing stage of the cyclone corresponding to the observations could also be accommodated there, see line 192). Moreover, I think it is worthwhile to discuss in further detail how the horizontal (given the spatial variability of the WV and ozone in the cyclone area) and

vertical resolution of MLS are adequate to the goals of the paper. 162-177 and fig. 2: I do not see the point to reproduce Fig.2, from Ravindra Babu et al., 2015, here. In 3.1 I do not see any novelty with respect to the analysis presented in that 2015 paper. The methodology and main results of that paper could be just shortly described and summarized. 207-209:How robust is this feature in the data? Are all cyclones contributing to such enhancement? 224 and 246: Cyclone winds can lose their axial symmetry near the top of the cyclone, and concentrate in one or two curved outflow jets. The authors may review the literature and see whether this can explain the upper level asymmetry in ozone and WV anomalies. 294: the authors should dwell more on the method they used to estimate the term Fam. At present, it seems their choice of 0.5 hPa is quite arbitrary. 299- 303: It seems this spatial asymmetry is a common, constant feature throughout the database ". . . the downward flux is always more. . ." . the authors should really dwell more on that, trying to find possible explanation in terms of the cyclone dynamics. 330: "intensify" for "intensity"? 330-339: It seems that (exactly) these results are already been reported in the quoted Ravidra Babu et al., 2015 paper. I do not understand why they are repeated here. 364: "intensity" for "intense" ? 366: "effecting" for "affecting"? Figure 1 caption, "strom" for "storm"

---

## Author Comment (AC1) · 12 Apr 2016

The paper presents the impact of cyclones that occurred over the North Indian Ocean during 2007-2013 on stratosphere-troposphere exchange using satellite measurements. Changes in ozone and water vapour distribution in the upper troposphere and lower stratosphere were analyzed. The cross-tropopause mass flux was estimated. The manuscript has some significant shortcomings. Therefore, I recommend some important revisions to address the comments listed below before publication by ACP.

Reply: First of all we wish to thank the reviewer for going through the manuscript

carefully and offering potential solutions to improve the manuscript content further.

General comments:

1) Scientific significance The paper presents new interesting results, however the results need to be better developed.

Reply: Thanks for appreciating actual content of the manuscript. We have revised the manuscript while considering both the reviewers comments/suggestions.

2) Scientific quality One important questions is whether the MLS measurements have sufficient spatial and temporal resolution to apply the used methodology? This has to be demonstrated. The explanation how the cross tropopause mass flux is calculated and which data are used is confusing. The method is explained in Sect. 2 and the used data are introduced in Sect. 3.1. I recommend to combine this in one Section. Further, the method of Ravindra Babu et al., 2015 is used (e.g. Fig. 2). However, the reader cannot understand this method without reading Babu et al., 2015. I recommend to provide more information about this method in Sect. 2. Many general statements have not been established with references (e.g. within the introduction, see below specific comments).

Reply: More details are provided in the revised manuscript with related to MLS data resolution, tropopause mass flux calculation and the methodology that is adapted from Ravindra Babu et al. (2015). We have not provided these details earlier to avoid repetition and/or plagiarism report. For MLS data resolution, first we separated MLS overpasses with respect to cyclone centre for each day of cyclone period and we made it cyclone-centre composite of corresponding ozone and water vapor, respectively. For tropopause mass flux, we considered whatever available tropopause temperature and pressure within 500km from the cyclone centre taken from Ravindra Babu et al. (2015) and winds within 500 km from the cyclone centre are taken from ERA-Interim data sets.

3) Presentation quality The presentation quality needs some improvements. There are

number of language and grammar issues. Further a lot of blank characters are missing, in particular after mathematical symbols or brackets. In the manuscript, abbreviations are still used that are not introduced. In some figures, the legend is missing.

Reply: We are sorry for the grammatical mistakes which have been reduced to the maximum possible extent in the revised manuscript. Missing of blank characters is mainly due to software problem loaded in one of our computers which is rectified now. We have elaborated all the abbreviations used in the manuscript when they appear for the first time in the manuscript. Specific comments:

1. Introduction: p. 3, line 51: 'Tropical cyclones with deep convective synoptic scale systems persisting for a few days to weeks play an important role on the mass exchange between troposphere and stratosphere and vice versa.' Please add some references.

Reply: Added.

p. 3, line 52: 'They transport large amount of water vapor, energy and momentum to the upper troposphere and lower stratosphere (UTLS) region.' Please add some references.

Reply: Added.

p. 3, line 60: 'The transport of water vapour and ozone around the tropopause caused by the cyclones can affect the radiation balance of the atmosphere.' Please add some references.'

Reply: Added.

p. 3, line 62: 'Increase of water vapor in the LS region will leads to a warming and ozone loss in this atmospheric region (Stenke and Grewe, 2005).' An increase of stratospheric water vapor contributes to tropospheric warming and stratospheric cooling, see e.g.: Climate Change 2007: The Physical Science Basis. Contribution of Working Group I to the Fourth Assessment Report of the Intergovernmental Panel on Climate Change, ed.

S. Solomon, D. Qin, M. Manning, M. Marquis, K. Averyt, M. M. B. Tignor, H. L. Miller and Z. Chen, Cambridge University Press, Cambridge, UK, and New York, NY, USA, 2007, pp. 1-996. D. Rind and P. Lonergan, J. Geophys. Res., 1995, 100, 7381-7396 P. Forster and K. P. Shine, Geophys. Res. Lett., 1999, 26, 3309-3312 V. L. Dvortsov and S. Solomon, J. Geophys. Res., 2001, 106, 7505-7514. D. T. Shindell, Geophys. Res. Lett., 2001, 28, 1551-1554. P. Forster and K. P. Shine, Geophys. Res. Lett., 2002, 29, 1086-1089. G. Myhre, S. J. Nilsen, L. Gulstad, K. P. Shine, B. Rognerud and I. S. A. Isaksen, Geophys. Res. Lett., 2007, 34, L01807. However, small changes of water vapor in the lower stratosphere have an impact on surface climate, see e.g: Riese et al., Impact of uncertainties in atmospheric mixing on simulated UTLS composition and related radiative effects, J. Geophys. Res., 117, D16305, doi:10.1029/2012JD017751, 2012. Solomon et al., Contributions of stratospheric water vapor to decadal changes in the rate of global warming, Science, 327, 1219-1223, 2010.

Reply: Thanks for updating us while providing above references. Most of the above mentioned references are included in the revised manuscript at appropriate places.

p. 3, line 65: troposphere air ! tropospheric air Reply: Corrected.

p. 4, line 82: 'TC event': abbreviation is not explained p. 4, line 86: 'MST Radar observations': abbreviation is not explained p. 4, line 87: 'BoB': abbreviation is not explained

Reply: These are explained in the revised manuscript.

p. 4, line 87: 'More literature related to influence of cyclones on the UTLS structure and composition is presented in Cairo et al. (2008).' Unspecific statement: please add some details or remove Cairo et al. 2008.

Reply: We have added major findings of Cairo et al. (2008) in the revised manuscript.

p. 5, line 105: 'COSMIC' is not explained Reply: Explained in the revised manuscript.

2. Data and Methodology p. 5, line 116: How many MLS profiles or measurements

(spatial and temporal resolution, horizontal distance between tracks) contribute to one typhoon event. Please add here some information and demonstrate that the data density is sufficient.

Reply: We have included details in the revised manuscript in the form of table (table 2).

p. 5, line 120: Which definition is used for the tropopause?

Reply: We used cold point and lapse rate tropopause definitions in this present study. For calculating tropopause mass flux, we used lapse rate tropopause definition only.

p. 6, line 135: Please add the precise time period for pre- and post-monsoon season and explain why you exclude the monsoon season.

Reply: Added in the revised manuscript. We also included monsoon season.

p. 7, line 149: 'tropopause parameters': Which parameters? Please combine this paragraph with details from Sect. 3.1'.

Reply: We combine and explained clearly this aspect in the revised manuscript.

3. Results and discussion p.7, line 162: How are the climatological mean values calculated? Is the monsoon season in the climatological mean excluded? During the Asian monsoon season the tropopause above the Asian monsoon anticyclone is elevated. Therefore, during this time period the lapse rate tropopause altitude differs from the altitude during the rest of the year. Is this considered in your analysis?

Reply: We have not considered this in the calculation. There could be day-to-day to the inter-annual variability in the observed climatological tropopause parameters. Since large data (14 years) have gone through climatology, we assume that variability less than the solar cycle is nullified, if not removed completely. Further Asian monsoon anticyclone aspect is related to the latitudes greater than 25oN, thus, do not affect our study in a significant manner. However, upper level anti-cyclonic circulation over the cyclones is reflected very well in our observations.

p. 7, line 169: How many measurements (tracks) do you have within 1000 km radius for one cyclone?

Reply: The total available RO measurements are not fixed for each cyclone; the RO measurements will change one cyclone to another. For example, the total RO measurements in the case of Nargis cyclone are 73. These details are provided in the revised manuscript.

p. 7, line 175: How is the cyclone intensity considered in the methodology of Ravindra Babu et. al, 2015? Please give a short summary about the method of Ravindra Babu et al., 2015 used for Fig. 2. How is vertical uplift at different flanks of the cyclone and difference between individual cyclones considered?

Reply: In Ravindra Babu et al. (2015), we did tropopause analysis based on different intensities of the cyclones such as depression (D), deep depression (DD), cyclonic storm(CS), severe cyclonic storm (SCS) and very severe cyclonic storm (VSCS). After detailed analysis we found that there is no major variation between D and DD, SC and SCS. So we combined the results of D and DD as one category and CS and SCS as another category and VSCS as one category. From each cyclone we separated the RO measurements based on the intensity and we combined.

p. 12, line 280-284: '...higher ozone mixing ratios are observed in the western and northwest side and more water vapor is located at the eastern side of the cyclonic center....' Why do you have this preference for the western and eastern side, respectively? In the schematic diagram Fig. 6 upward and downward transport of water vapor and ozone is shown. The diagram implies rotational symmetry around the center of the cyclone. How fits the rotational symmetry together with the preference at the western and eastern side?

Reply: Our results from Ravindra babu et al. (2015) shows the integrated RH is more in the east and south east side within 500 km from the cyclone centre and the COH, TTL thickness also shows high in the north and north west side within 500 km from

the cyclone centre. From these we assume that different sides within 500 km from the centre there may be different variations in the ozone and water vapour as well as cross tropopause flux. That's why we calculated the flux with respect to sector wise from the cyclone centre. The diagram shown in the figure 6 it is just assumption of the cyclone structure only. Our main aim of the figure 6 is to show the variation of tropopause parameters in the schematic way i.e., ozone coming down from the lower stratosphere due to subsidence at the centre and water vapour entering in to the lower stratosphere due to anti-cyclonic circulation above the cyclone. The higher ozone mixing ratios are observed in the western and northwest side and more water vapour is located at the eastern side of the cyclone centre because of the upper level anti-cyclonic circulation over the cyclones. This will push the water vapour towards the south and east side of the cyclone centre. In the other side of the cyclone, the detrainment of the lower stratospheric air may occur along with strong subsidence in the cyclone centre. This might be the region for higher ozone in the west and northwest side and more water vapour in the east and southeast side of the cyclone centre. Note that Ray and Rosenlof (2007) also reported higher water vapour mixing ratios in the east side of the cyclone centre for Atlantic and Pacific oceans. Further, very recently Reutter et al. (2015) reported that the more stratosphere- troposphere transport takes place in the west side of the cyclone centre due to west ward tilt of the cyclone with height.

p. 12, line 294: 'by assuming change in the tropopause pressure by 0.5 hPa' Why 0.5 hPa is used?

Reply: Since we do not have pressure variation with time we have assumed different pressures while considering minimum to maximum possible pressure variations.

p. 13, line 299: Please explain why different cross-tropopause flux occurs in different sectors.

Reply: As we found different variations in the water vapour and ozone transport in

different sectors, we have estimated cross-tropopause flux for these different sectors. Please see reply for above comment (p. 12, line 280-284) for more details.

4. Summary and conclusions p. 14, line 335: 'The main findings of the present communication are summarized below.' ! Our main findings are summarized below.'

Reply: Modified.

p. 14, line 336-339: 'Lowering of CPH (0.6 km) and LRH (0.4 km) values with coldest CPT and LRT (2-3K) within a 500 km radius from the cyclone centre is noticed. Higher (2 km) COH leading to the lowering of TTL thickness ( 3 km) is clearly observed (Ravindra Babu et al., 2015).' That is a result from Ravindra Babu et al, 2015 and not from the present paper Ratman et al.. That should be clearly recognizable in the text.

Reply: We have already provided reference when it is mentioned.

p. 15, line 346-347: 'Interestingly significant enhancement in the lower stratosphere (82 hPa) water vapor is noticed in the east and SE side from the cyclone centre.' Again, why only at the east and SE side?

Reply: Please see explanation provided for the comment p. 12, line 280-284.

p. 15, line 355-357: 'Strong convective towers with strong updrafts extending up to the tropopause altitude in the form of spiral bands extending from 500 to 1000 km are present.' In Fig. 6, three bands of downward transport of ozone and three bands for upward transport of water vapor are drawn which are not visible in Fig. 3 and 4. Please explain this discrepancy or adapt Fig. 6. To confirm the spiral bands of upward and downward transport illustrated in Fig. 6 trajectory calculations would be very helpful.

Reply: Note that figure 3 and 4 are cyclone-centre composite of ozone and water vapour obtained from all 16 cyclones and the figure 6 is the only schematic picture of a cyclone. Our main aim in figure 6 is to show the variation of tropopause parameters in the form of schematic way i.e., ozone coming down at the centre from the lower stratosphere due to subsidence and water vapour entering in to the lower stratosphere

due to anti-cyclonic circulation above the cyclone above the spiral bands.

Figures: Fig. 1: 'strom' ! 'storm' Fig. 3: Legend from a-d is missing. Fig. 4: Legend from a-d is missing.

Reply: Corrected in the revised manuscript.

—END—

---

## Author Comment (AC2) · 12 Apr 2016

This is an interesting study of the impact of cyclones on ozone and water vapour in the upper troposphere and lower stratosphere. It is based on the analysis of some cases study, using satellite measurements to estimate the air flux across the tropopause. It is surely a valuable contribution on a hot topic in stratospheric research, since the ability to predict future changes in the stratosphere relies on correct estimates on how tropical troposphere to stratosphere transport might evolve. I definitely agree that the role of deep convection in cyclones is worth of more research, and it is appropriate for the

journal, so I encourage the publication of this work. However, there are a number of open issues that have to be addressed; therefore I recommend a revision before publication. I had the chance to read the general comments of the Anonymous Reviewer #2 and I do share all his/her general comments.

Reply: First of all we wish to thank the reviewer for going through the manuscript carefully, appreciating actual content of the manuscript and offering potential solutions to improve the manuscript content further. We have revised the manuscript while considering both the reviewers comments/suggestions.

In particular I find strange how the results from previous work of Ravindra Babu et al. (2015) are used in the present paper: on one hand, figures and conclusions from that paper are reproduced in a way that seems redundant, on the other hand a description of the method used in that work, which is duplicated in the present one, is lacking so to force the reader to go to the original reference. I therefore suggest to briefly summarize the results AND methods presented in Ravindra Babu et al., and to skip fig.2.

Reply: The methodology explained in Ravindra Babu et al. (2015) is re-produced briefly in the current manuscript as suggested. Note that figure 2 is very important even for the current manuscript and thus retained.

Detailed comments:

lines 49-52: These sentences seems more to describe what the article is aimed for, than an introduction, The authors should support their claims with references, or the sentences should be made less assertive.

Reply: We have provided more references at the appropriate places as also mentioned by other reviewer.

61: Again, the assessment of the effectiveness of cyclones in promoting STE is the objective of the paper. References should be made to previous studies supporting this claim, or the sentence should be dropped, or reformulated to introduce the aim of the

paper.

Reply: We have added relevant references for the text used in the present study at appropriate places.

62-63: The Stenke and Grewe paper deals mainly with the impact of water vapour increase on ozone chemistry. I did not find any claim of temperature increase induced by an increase of WV, there. On the contrary, there is a lot of modeling evidence (and even some experimental study, see as instance Maycock et al., Q. J. R. Meteorol. Soc., 2014), in the literature, that an increase stratospheric water vapor would lead to a cooling of stratospheric temperatures. So the sentence in the paper seems not correct.

Reply: We have corrected the sentence while adding suitable reference.

82, 86, 87,96: TC, MST, BoB, COSMIC, abbreviations have not been introduced earlier.

Reply: The abbreviations are elaborated when they appear for the first time in the revised manuscript.

91: The findings presented in Cairo et al. (2008) should be reported.

Reply: Reported.

128-134: Such information should be presented as a table.

Reply: We have added one more table with the classification of cyclones over north Indian Ocean as suggested.

139-140: This sentences is not clear. Is it suggesting that only long lasting cyclones have been selected in order to have enough MLS WV profiles in the cyclone area? This is quite an important point, and the average number of MLS profiles used should be quoted, maybe even in the form of a table, for each cyclone (the developing stage of the cyclone corresponding to the observations could also be accommodated there, see line 192). Moreover, I think it is worthwhile to discuss in further detail how the horizontal (given the spatial variability of the WV and ozone in the cyclone area) and

vertical resolution of MLS are adequate to the goals of the paper.

Reply: We reported available MLS profile for each cyclone in the form of table in the revised manuscript.

162-177 and fig. 2: I do not see the point to reproduce Fig.2, from Ravindra Babu et al. (2015), here. In 3.1 I do not see any novelty with respect to the analysis presented in that 2015 paper. The methodology and main results of that paper could be just shortly described and summarized.

Reply: It is well known that the tropopause characteristics play and important role in controlling the STE processes. Though the tropopause characteristics are mentioned in our earlier draft, we would like to retain figure 2 in this paper as it will be easy to refer the tropopause characteristics by the readers so that this paper will remain stand alone. This will also avoid going through our earlier paper as rightly mentioned by both the reviewers.

207-209: How robust is this feature in the data? Are all cyclones contributing to such enhancement?

Reply: It will change based on cyclone intensity. This will be more in the case of maximum intensity of cyclone such as SCS and VSCS category. Please see figure 5 for more details. Note that we calculated based on intensity and are not showed in the manuscript. However, our analysis confirms that the ozone is more in the case of VSCS compare other SCS and DD categories. During the VSCS time the ozone detrainment is reached to the 146 hPa level. Since the available profiles of MLS are less for different intensities so we combined all the profiles that are available within 1000 km from the centre of all 16 cyclones.

224 and 246: Cyclone winds can lose their axial symmetry near the top of the cyclone, and concentrate in one or two curved outflow jets. The authors may review the literature and see whether this can explain the upper level asymmetry in ozone and WV

anomalies.

Reply: This is very important point that the cyclone winds play important role in the distribution of the water vapour and ozone above the cyclone. As mentioned earlier, the higher ozone mixing ratios are observed in the western and northwest side and more water vapour is located at the eastern side of the cyclone centre because of the upper level anti-cyclonic circulation over the cyclones. This will push the water vapour towards the south and east side of the cyclone centre. In the other side of the cyclone, the detrainment of the lower stratospheric air may occur along with strong subsidence in the cyclone centre. This might be the region for higher ozone in the west and northwest side and more water vapour in the east and southeast side of the cyclone centre. Note that Ray and Rosenlof (2007) also reported higher water vapour mixing ratios in the east side of the cyclone centre for Atlantic and Pacific oceans. Further, very recently Reutter et al. (2015) reported that the more stratosphere-troposphere transport takes place in the west side of the cyclone centre due to west ward tilt of the cyclone with height. These aspects are mentioned in the revised manuscript.

294: the authors should dwell more on the method they used to estimate the term Fam. At present, it seems their choice of 0.5 hPa is quite arbitrary.

Reply: Since we do not have pressure variation with time we have assumed different pressures while considering minimum to maximum possible pressure variations, which is the best way when no observations are present.

299- 303: It seems this spatial asymmetry is a common, constant feature throughout the database ". . . the downward flux is always more. . ." . the authors should really dwell more on that, trying to find possible explanation in terms of the cyclone dynamics.

Reply: The tropopause flux is calculated for each cyclone maximum intensity day only so on the higher intensity time within 500 km from the cyclone centre the anti-cyclonic flow dominated and cause the upward flux in the east and southeast side. Whereas, subsidence dominating in the other side cause downward flux in the west and north-

west side of the cyclone.

330: "intensify" for "intensity"? 330-339: It seems that (exactly) these results are already been reported in the quoted Ravidra Babu et al., 2015 paper. I do not understand why they are repeated here.

Reply: For completeness we have included these sentences in this paper also as someone may be interested to see the tropopause variations during these cyclones and to make this manuscript standalone we retained those statements and related figure.

364: "intensity" for "intense" ? 366: "effecting" for "affecting"? Figure 1 caption, "strom" for "storm"

Reply: Corrected in the revised manuscript.

—END—

---

## Referee Report (RR1)

**Reviewer Comment**

**10.5194/acp-2015-988 (Editor - Rolf Müller) Atmos. Chem. Phys. Discuss.**
**'Effect of tropical cyclones on the Stratosphere-Troposphere Exchange observed using satellite observations over north Indian Ocean'**
* * *
**General comments:**

The paper is much improved compared to the previous ACPD version. Unfortunately, the submitted version of the paper with 'change track' does not show the changes related to the ACPD Version. Therefore, it is very difficult to identify all the changes. The manuscript has still some shortcomings. Therefore, I recommend some revisions to address the comments listed below before publication by ACP.

Presentation quality
The presentation quality is better than in the last version, however there are still plenty of redundant blank characters and some grammar issues. I recommend careful proof-reading by the production department of ACP.

**Specific comments:**

1. Introduction:

p. 3, line 55: remove blank character 'water vapor- poor' or use 'dry'

p. 3, line 65: 'Increase of water vapor in the LS region will leads to troposphere warming and stratospheric cooling might be due to lose ozone ...'
Water has major consequences for the radiative balance and heat transport in the atmosphere. Enhanced ozone loss is a secondary effect of increasing water vapor.

p. 3, line 67: remove blank character '... Change, 2007) . Even ...'

p. 3, line 67: 'Solomon et al. (2010) reported the relation between global warming and lower stratospheric water vapor.'
Unspecific statement: please add some details

p. 3, line 71: 'long term' to 'long-term' (?)

p. 5, line 101-102: '..and also reported on the impact of the TCs on in the UTLS region on the regional scales.'
What does this mean?

p. 5, line 103: insert blank character 'RavindraBabu'

p. 5, line 114: remove blank character 'Aura- Microwave'

2. Data and Methodology

p. 5, line 119: remove blank character 'Aura -MLS'

p. 6, line 127: remove '(1)' ??

p. 6, line 129: What is the meaning of 'best track data'?

p. 6, line 140: remove 'the' before 'Table 2'

p. 7, line 149: 'We have $94 \pm 21$ mean MLS profiles for each cyclone' That means you use all MLS profiles for one cyclone (e.g. O3B) for all days of the cyclone period (4days) within $1000\,\mathrm{km}$ from the cyclone center. Is that correct? A figure showing the position of the MLS tracks (profiles) in the cyclone-centered coordinate system would be very helpful to see the data coverage.

p. 7, line 164: 'Large convection around the eye and ...'
Please add 'eye of the cyclone'

3. Results and discussion

Just for understanding: Figure 2 is for the period 2002-2013 and Figure 3 for 2007-2013. Why do you use different time periods?

Compared to the previous version of the manuscript cyclones during the monsoon season (03B, PHEt, Gonu) are in addition included in the new version of the paper (20% of all profiles). Therefore, you use a different set of MLS data for calculating Fig. 3 and 4 in the new and old version of the paper. Is that correct? I am wondering why Figure 3 and 4 are exactly the same in the new and old version of the paper. I would expect some differences.

4. Summary and conclusions

I still recommend to adapt Fig. 6. The diagram does not describe the east-west asymmetry found in Figs- 3-5. From Fig.3, it is not clear that upward transport of high ozone values from the lower stratosphere into the troposphere occurs outside of the cyclone center (red arrows) as shown in Fig. 6.

Figures and Tables:

Table 2: Please add time unit for column 'total sustained' and 'sub-stained time with maximum intensity'. The year (or date) of the cyclone occurrence would be in addition a useful information.

---

## Referee Report (RR2)

**Reviewer Comment**

**10.5194/acp-2015-988 (Editor - Rolf Müller) Atmos. Chem. Phys. Discuss.**
**'Effect of tropical cyclones on the Stratosphere-Troposphere Exchange observed using satellite observations over north Indian Ocean'**
* * *
**General comments:**

The paper is much better elaborated as before, however there are two important issues that I recommend to improve.

**1) Figure 7:**

You wrote:*Kindly note that the diagram shown in the figure 6 (now figure 7) is a typical structure of the cyclone but not exactly what we observed in the Figures 3 or 4.*

In that case some references related to Fig. 7 (e.g. adapted from ...) should be added which show the typical structure of a cyclone. Further, the differences between your results and the typical structure of a cyclone have to be discussed.

**2) MLS Data for figures 4 and 5**

In your first comment you wrote on page 3, line 116-118:

p. 6, line 135: Please add the precise time period for pre- and post-monsoon season and explain why you exclude the monsoon season.
*Reply: Added in the revised manuscript. We also included monsoon season.*

In your second comment your wrote on page 1, line 27-34

Further, you have added more observations and you have nicely provided background information in tables. However, the central figures 3 and 4 seem

unchanged. Is this true? Or do you have to update the figures after the inclusion of the new data?

*Reply: Kindly note that we have used same data set in the current version of the manuscript. Please go through Table 1 where we have mentioned all the cyclones during our initial submission itself. Thus, there is no change in the data set used in the current version and old version. Only difference is we show them according to the season in the revised manuscript.*

If I understand you right, the monsoon cases were already used in the Figures 4 and 5 of the ACPD version and only the text was imprecisely formulated in the ACPD version. In the ACPD version, only tropical cyclones in the pre- and post-monsoon season were mentioned in the text suggesting that tropical cyclones during monsoon season are not included in Fig. 4 and 5. Therefore I am wondering that Figure 4 and 5 are the same because I thought in the ACPD version tropical cyclones during monsoon season are not considered. Please clarify this issue.

---

## Author Response (AR2)

**Replies to Editor comments/suggestions**

Thanks for your revised version, which now has been seen by the two original referees again. One referee is satisfied with your changes, but the second referee (while agreeing that you have made substantial progress) still has important issues with your manuscript. Therefore I would ask you to revise the paper again taking the comments by the referee into account.

**Reply: We wish to thank editor for going through the manuscript and allowing us to revise again for the betterment. We have taken care of all the suggestions made by the reviewer.**

A few comments also from my side: The major remaining point seems to be the message conveyed by Figure 6 and the supporting evidence from the MLS observations (which have a limited spatial resolution). For example, in Fig. 6 you show 5 downward arrays but is this structure really reflected in Figs. 3 and 5?

**Reply: Kindly note that the diagram shown in the figure 6 (now figure 7) is a typical structure of the cyclone but not exactly what we observed in the Figures 3 or 4. Figures 3 and 4 show the climatology of cyclone centred composite of the Ozone and Water vapour obtained from 16 cyclones. Our main interest to show a schematic picture is to convey the message on how the tropopause parameters vary and from where the mass flux will occur. Ozone comes down from the lower stratosphere due to subsidence at the centre of the cyclone and water vapour enters in to the lower stratosphere from the side bands due to anti-cyclonic circulation. Up and down arrows show the updrafts and downdrafts, where we can regularly observe in the cyclone structure.**

**There was one mistake i.e., representing the ozone coming from the side ways of the cyclone, which is rectified in the revised manuscript.**

Further, you have added more observations and you have nicely provided background information in tables. However, the central figures 3 and 4 seem unchanged. Is this true? Or do you have to update the figures after the inclusion of the new data?

**Reply: Kindly note that we have used same data set in the current version of the manuscript. Please go through Table 1 where we have mentioned all the cyclones during our initial submission itself. Thus, there is no change in the data set used in the current version and old version. Only difference is we show them according to the season in the revised manuscript.**

Finally, there are a lot of technical/typesetting issues to be resolved in the revised version, as listed in the review. I have also noted that there is a typo on page 13, "Wie" should be "Wei" and the quantity "F" should be in italics in the text. Please also abbreviate the journal in reference Wei (1987).

**Reply: We wish to inform that there was some problem in one of our computers (MS word) which is creating additional space. In the revised version of the manuscript, we have taken care of these typos. If this problem still persists, we will take help of production department of ACP. We have rectified the mistake in reference mentioned above.**

In the revised version please make sure that the changes made are clearly noticeable (see also review).

**Reply: Kindly note that we have incorporated all the suggestion made by the reviewer earlier with track changes and was uploaded. Our sincerely apology for not able to see the changes and we don't know what happened exactly.**

**Replies to Anonymous Referee #2 comments/suggestions**

General comments:

The paper is much improved compared to the previous ACPD version. Unfortunately, the submitted version of the paper with 'change track' does not show the changes related to the ACPD Version. Therefore, it is very difficult to identify all the changes. The manuscript has still some shortcomings. Therefore, I recommend some revisions to address the comments listed below before publication by ACP.

**Reply: The authors wish to thank the reviewer for his/her thorough review of the manuscript and for offering recommendations to improve the manuscript content. We have provided point-by-point replies to the reviewer's comments and information on how we have handled the revised manuscript. Kindly note that we have incorporated all the suggestion made by the reviewer earlier with track changes and was uploaded. Our sincerely apology for not able to see the changes and we don't know what happened exactly.**

Presentation quality

The presentation quality is better than in the last version, however there are still plenty of redundant blank characters and some grammar issues. I recommend careful proof-reading by the production department of ACP.

**Reply: Thank you very much for your appreciation. We wish to inform that there was some problem in one of our computers (MS word) which is creating additional space. In the revised version of the manuscript, we have taken care of these typos. If this problem still persists, we will take help of production department of ACP.**

Specific comments:

**1. Introduction:**

p. 3, line 55: remove blank character 'water vapor- poor' or use 'dry'

**Reply: Removed.**

p. 3, line 65: 'Increase of water vapor in the LS region will leads to troposphere warming and stratospheric cooling might be due to lose ozone ...' Water has major consequences for the radiative balance and heat transport in the atmosphere. Enhanced ozone loss is a secondary effect of increasing water vapor.

**Reply: We have changed this sentence in the revised manuscript as suggested.**

p. 3, line 67: remove blank character '... Change, 2007) . Even ...'

**Reply: Removed.**

p. 3, line 67: 'Solomon et al. (2010) reported the relation between global warming and lower stratospheric water vapor.' Unspecific statement: please add some details

**Reply: We have revised this statement with better clarity in the revised manuscript.**

p. 3, line 71: 'long term' to 'long-term' (?)

**Reply: Corrected.**

p. 5, line 101-102: '..and also reported on the impact of the TCs on in the UTLS region on the regional scales.' What does this mean?

**Reply: We have re-written this statement with better clarity.**

p. 5, line 103: insert blank character 'RavindraBabu'
p. 5, line 114: remove blank character 'Aura- Microwave'
**Reply: Removed.**
**2. Data and Methodology**
p. 5, line 119: remove blank character 'Aura -MLS'
p. 6, line 127: remove '(1)' ??
**Reply: Removed.**
p. 6, line 129: What is the meaning of 'best track data'?
**Reply: This is the IMD observed tropical cyclones best track data. We changed it in the**
**revised manuscript.**
p. 6, line 140: remove 'the' before 'Table 2'
**Reply: Removed.**
p. 7, line 149: 'We have $94 \pm 21$ mean MLS profiles for each cyclone' That means you use
all MLS profiles for one cyclone (e.g. O3B) for all days of the cyclone period (4days) within
1000 km from the cyclone center. Is that correct? A figure showing the position of the MLS
tracks (profiles) in the cyclone-centered coordinate system would be very helpful to see the
data coverage.
**Reply: Yes, we have used all MLS profiles during cyclone period (all cyclone days). The**
**MLS overpasses have been separated with respect to cyclone centre for each day of the**
**individual cyclone. As suggested by the reviewer, we added one more figure (as figure 2)**
**to show the total MLS profiles around the cyclone–center used in the present study.**
p. 7, line 164: 'Large convection around the eye and ...' Please add 'eye of the cyclone'
**Reply: Added.**
**3. Results and discussion**
Just for understanding: Figure 2 is for the period 2002-2013 and Figure 3 for 2007-2013.
Why do you use different time periods?
**Reply: Figure 2 shows the Cyclone centered – composite of mean difference in the**
**tropopause parameters between climatological mean obtained by using GPS RO data**
**from the year 2002-2013 and individual tropopause parameters observed during**
**cyclones that have occurred during 2007-2013 respectively.**
**In general, if we want to represent any parameter climatologically, data length should**
**contain at least one solar cycle so that while making composite, all the dominant**
**oscillations like SAO, AO, QBO, ENSO, solar cycle etc., will be removed and remaining**
**will represent true background. Thus, we make use of all GPS RO data available since**
**2002 (11 years). We also checked by considering data only during 2007-2013 as a**
**background but could not see any major difference. This is mainly because of less**
**number of GPS RO data available before 2007.**
Compared to the previous version of the manuscript cyclones during the monsoon season
(03B, PHEt, Gonu) are in addition included in the new version of the paper (20% of all
profiles). Therefore, you use a different set of MLS data for calculating Fig. 3 and 4 in the
new and old version of the paper. Is that correct? I am wondering why Figure 3 and 4 are
exactly the same in the new and old version of the paper. I would expect some differences.

**Reply: We have used same data set in the current version of the manuscript. Please go through Table 1 where we have mentioned all the cyclones during our initial submission itself. Thus, there is no change in the data set used in the current version and old version. Only thing is we show them according to the season in the revised manuscript.**

**4. Summary and conclusions**

I still recommend to adapt Fig. 6. The diagram does not describe the eastwest asymmetry found in Figs- 3-5. From Fig.3, it is not clear that upward transport of high ozone values from the lower stratosphere into the troposphere occurs outside of the cyclone center (red arrows) as shown in Fig. 6.

**Reply: It is our mistake to represent the ozone coming from the side ways of the cyclone which is rectified in the revised manuscript. Please note that the diagram shown in the figure 6 (now figure 7) is a typical structure of the cyclone but not exactly what we observed in the Figures 3 or 4. Figures 3 and 4 show the climatology of cyclone centred composite of the Ozone and Water vapour obtained from 16 cyclones. Our main interest to show a schematic picture is to convey the message on how the tropopause parameters vary and from where the mass flux will occur. Ozone comes down from the lower stratosphere due to subsidence at the centre of the cyclone and water vapour enters in to the lower stratosphere from the side bands due to anti-cyclonic circulation. Up and down arrows show the updrafts and downdrafts, where we can regularly observe in the cyclone structure.**

Figures and Tables:

Table 2: Please add time unit for column 'total sustained' and 'sub-stained time with maximum intensity'. The year (or date) of the cyclone occurrence would be in addition a useful information.

**Reply: We have added time units and year of the cyclones occurred as suggested in the revised manuscript.**

**We once again thank the reviewer for going through the manuscript carefully and offering potential solutions to further improve the manuscript content.**

---END---

[revised manuscript text omitted]

---

## Author Response (AR3)

**Replies to Reviser comments/suggestions** 1 2 3 4 General comments: The paper is much better elaborated as before, however there are two 5 important issues that I recommend to improve. Reply: The authors wish to thank the reviewer for going through the manuscript again 6 and offering very precise comments/suggestions. We have provided detailed explanation 7 8 to the both the issues raised by the reviewer. 9 1) Figure 7: You wrote: Kindly note that the diagram shown in the figure 6 (now figure 7) is 10 11 a typical structure of the cyclone but not exactly what we observed in the Figures 3 or 4. 12 In that case some references related to Fig. 7 (e.g. adapted from ...) should be added which show the typical structure of a cyclone. Further, the differences between your results and the 13 typical structure of a cyclone have to be discussed. 14 Reply: Thank you very much for the suggestion. We have quoted suitable reference 15 from where we have taken basic idea of this schematic figure as suggested. The below 16**

- of
- 17 figure from chapter 9 figure is 6 18
- 19

A' н Evewall Descending air current Ascending air curre Sea surface

Elements of a severe tropical cyclone. A-A- is schematic. NOAA photograph.

20 This is the NOAA photograph of typical structure of a tropical cyclone with updrafts 21 22 and downdrafts. The top panel is taken as schematic to show the cyclone structure with 23 additional tropopause parameters. Kindly note that the results presented in figure 4 and figure 5 is composite picture of all 16 cyclones. Therefore, it is to be noted that the 24 25 structure of tropical cyclone is not similar in all the cases.

- 26
- 27 2) MLS Data for figures 4 and 5
- 28 In your first comment you wrote on page 3, line 116-118:

- p. 6, line 135: Please add the precise time period for pre- and post-monsoon season andexplain why you exclude the monsoon season.
- 31 *Reply: Added in the revised manuscript. We also included monsoon season.*
- 32 In your second comment your wrote on page 1, line 27-34
- Further, you have added more observations and you have nicely provided background
- 34 information in tables. However, the central figures 3 and 4 seem unchanged. Is this true? Or 35 do you have to update the figures after the inclusion of the new data?
- 36 Reply: Kindly note that we have used same data set in the current version of the manuscript.
- Please go through Table 1 where we have mentioned all the cyclones during our initial
  submission itself. Thus, there is no change in the data set used in the current version and old
- 39 version. Only difference is we show them according to the season in the revised manuscript.
- 40 If I understand you right, the monsoon cases were already used in the Figures 4 and 5 of the
- ACPD version and only the text was imprecisely formulated in the ACPD version. In the
   ACPD version, only tropical cyclones in the pre and post-monsoon season were mentioned in
- 43 the text suggesting that tropical cyclones during monsoon season are not included in Fig. 4
- and 5. Therefore I am wondering that Figure 4 and 5 are the same because I thought in the
   ACPD version tropical cyclones during monsoon season are not considered. Please clarify
- 46 this issue.
- 47 Reply: We are sorry for the confusion. In our first version of the manuscript in ACPD,
  48 we have included the tropical cyclones (TCs) that are formed during monsoon months
- 48 we have included the tropical cyclones (TCs) that are formed during monsoon months 49 also in the name of '*Pre-Monsoon*'. In the revised version we have segregated all the 16
- 50 TCs according with season. We request the revised version we have segregated an the ro
- 51 earlier ACPD version entitled as 'Table 1. Cyclone name, cyclone Intensity (CI), centre
- 52 latitude, centre longitude, estimated central pressure and estimated cross-tropopause mass
- 53 flux with respect to cyclone centre for C1 (NW side), C2 (NE side), C3 (SW side) and C4
- 54 (SE side), respectively'. We have clearly mentioned about the TCs that are occurred 55 during monsoon season (03B, Gonu and Phet). In first version, we added the precise
- time for only pre-monsoon and post monsoon seasons. In the revised version, we have provided the precise time period for each season, therefore we mentioned in the replies as the monsoon season also included.
- Thus, no additional data has been included in the revised version and thus there will not be any change in the figure 4 and 5. The only difference between first version and second version of the manuscript is segregating the same number of 16 cases according to the season wise but number of cases i.e., 16 cyclones remain same. Hope we have clarified this issue now.
- We once again thank the reviewer for going through the manuscript carefully and
  brining out to our notice the errors/mistakes for improving the manuscript content
  significantly.
- 68

- 69
- 70

---END----

[revised manuscript text omitted]

| Cyclone | CI   | Centre
Latitude | Centre
Longitude | Estimated Central
Pressure (hPa) | Cl       | C2    | C3     | C4     |
|---------|------|--------------------|---------------------|-------------------------------------|----------|-------|--------|--------|
| 03B     | CS   | 23.5               | 66                  | 986 (25Jun2007)                     | -0.013   | 0.661 | -0.603 | -0.258 |
| Aila    | SCS  | 22                 | 88                  | 968 (25May2009)                     | 1.90E-04 | 0.191 | -0.299 | -0.072 |
| Helen   | SCS  | 16.1               | 82.7                | 990 (21Nov2013)                     | 0.025    | 0.216 | -0.095 | -0.11  |
| Jal     | SCS  | 11                 | 84                  | 988(6Nov2010)                       | 0.025    | 0.384 | -0.4   | -0.218 |
| Laila   | SCS  | 14.5               | 81                  | 986 (19May2010)                     | -0.012   | 0.123 | -0.352 | -0.299 |
| Mahasen | CS   | 18.5               | 88.5                | 990 (15May2013)                     | -0.006   | 0.354 | -0.473 | -0.256 |
| Nilam   | CS   | 11.5               | 81                  | 990 (31Oct2012)                     | 0.016    | 0.313 | -0.274 | -0.097 |
| Nargis  | VSCS | 16                 | 94                  | 962 (2May2008)                      | -0.828   | 0.094 | -1.946 | 0.384  |
| Giri    | VSCS | 19.8               | 93.5                | 950 (22Oct2010)                     | -0.518   | 0.022 | -0.823 | 0.032  |
| Gonu    | SuCS | 20                 | 64                  | 920 (4Jun2007)                      | -0.502   | 0.123 | -2.563 | 0.37   |
| Lehar   | VSCS | 13.2               | 87.5                | 980 (26Nov2013)                     | -0.55    | 0.119 | -2.019 | 0.411  |
| Madi    | VSCS | 13.4               | 84.7                | 986 (10Dec2013)                     | -0.375   | 0.054 | -1.449 | 0.352  |
| Phailin | VSCS | 18.1               | 85.7                | 940 (11Oct2013)                     | -0.9     | 0.179 | -2.576 | 0.479  |
| Phet    | VSCS | 18                 | 60.5                | 964 (2Jun2010)                      | -1.058   | 0.203 | -2.698 | 0.559  |
| SIDR    | VSCS | 19.5               | 89                  | 944 (15Nov2007)                     | -0.493   | 0.066 | -0.926 | 0.231  |
| Thane   | VSCS | 11.8               | 80.6                | 970 (29Dec2011)                     | -1.272   | 0.356 | -2.979 | 0.558  |

**703 Figures:**

Figure 1. Tropical cyclone tracks of different categories (cyclonic storm (CS, blue color),
severe cyclonic storm (SCS, orange color), very severe cyclonic storm (VSCS, red color)
and super cyclonic storm (SuCs, magenta color)) that occurred over North Indian Ocean
during 2007 - 2013.

---

## Author Response (AR4)

**Replies to Co-editor comments/suggestions**

Co-editor Decision: Publish subject to technical corrections (27 Jun 2016) by Dr. Rolf Müller
Comments to the Author: Thank you very much for the changes to the manuscript and the clarifications. I accept the manuscript for ACP (see some minor technical corrections below). Congratulations.
Reply: We thank you very much for getting through review and appreciating the actual content of the work. Though it has taken several iterations but we are happy to note significant improvement in the manuscript content from its first version to this stage.

l 468: change to: ... is a composite picture...
Reply: Corrected.

468/469: replace this sentence by: "Because each individual typhoon is different, this composite picture will differ somewhat from the typical structure shown in Fig. 7.
Reply: Replaced.

---END---

[revised manuscript text omitted]